# D3AD: Dynamic Denoising Diffusion Probabilistic Model for Anomaly Detection

## Abstract

Diffusion models have found valuable applications in anomaly detection by capturing the nominal data distribution and identifying anomalies via reconstruction. Despite their merits, they struggle to localize anomalies of varying scales, especially larger anomalies like entire missing components. Addressing this, we present a novel framework that enhances the capability of diffusion models, by extending the previous introduced implicit conditioning approach Meng et al. (2022) in three significant ways. First, we incorporate a dynamic step size computation that allows for variable noising steps in the forward process guided by an initial anomaly prediction. Second, we demonstrate that denoising an only scaled input, without any added noise, outperforms conventional denoising process. Third, we project images in a latent space to abstract away from fine details that interfere with reconstruction of large missing components. Additionally, we propose a fine-tuning mechanism that facilitates the model to effectively grasp the nuances of the target domain. Our method undergoes rigorous evaluation on two prominent anomaly detection datasets VISA and BTAD, yielding state-of-the-art performance. Importantly, our framework effectively localizes anomalies regardless of their scale, marking a pivotal advancement in diffusion-based anomaly detection. All code will be made public upon acceptance.

## 1 Introduction

Anomaly detection (AD) and related tasks such as identifying out-of-distribution data and detecting novel patterns, holds significant importance within the industrial sector. Applications range from detecting component defects Roth et al. (2022); Zou et al. (2022) and fraudulent activities Ahmed et al. (2016) to assistance in medical diagnoses Baur et al. (2019); Wyatt et al. (2022) through identification of diseases. Overlooked anomalies in these applications could result in adverse financial and safety repercussions. In the manufacturing sector, flawed components which remain undetected lead to high scrap costs or customer complaints. Moreover, manual inspection of defects is a laborious task which often results in visual strain, especially when assessing reflective parts repeatedly. Motivated by these challenges, we explore the intricacies of visual anomaly detection within industrial contexts. In computer vision, anomaly detection entails both classifying images as anomalous or normal and segmenting/localizing anomalous regions.

Typically, due to the scarcity of abnormal samples, an unsupervised approach is often employed for AD whereby a one-class classifier is trained on only nominal data. Such approaches can be grouped into representation-based and reconstruction-based methods. The latter reconstructs an anomalous input image, which is anomaly-free since the model is only trained on nominal data; thereby anomalies can be detected by simple comparison of the input with it's reconstruction. However, previous generative models Bergmann et al. (2019c); Gong et al. (2019) are easily biased towards the flawed input image leading to a reconstruction with the anomaly or artifacts. Diffusion models Sohl-Dickstein et al. (2015); Ho et al. (2020) have shown success in image and video synthesis Nichol et al. (2022); Rombach et al. (2022); Blattmann et al. (2023), 3D reconstruction Poole et al. (2023), music generation Kong et al. (2021) etc. They have also been used for the AD task acquiring promising results Wyatt et al. (2022); Mousakhan et al. (2023) but their full potential in anomaly detection remain untapped.

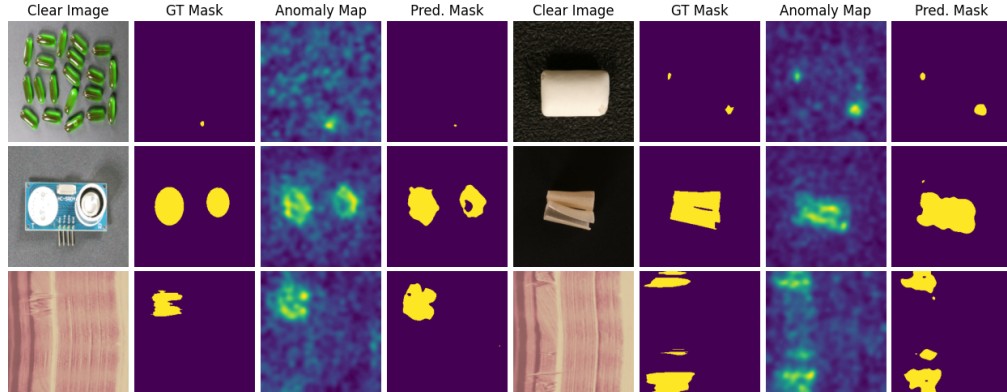

Figure 1: D3AD segmentation results of anomalies across scales from VisA and BTAD.

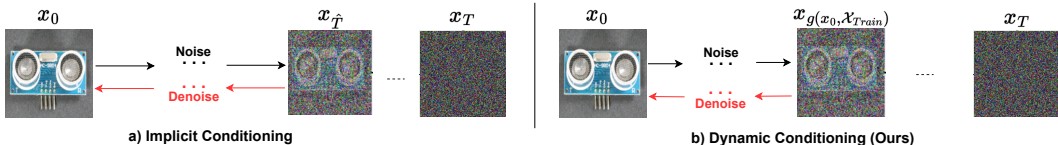

Figure 2: Dynamic conditioning whereby the amount of added noise is a function of the input image and training dataset dependent on an initial guess of the severity of the anomaly.

Anomalies occur in diverse forms from small scratches to complete missing components, see Figure 1. In previous AD diffusion models, we observe that simple application of fixed noise to an anomalous input image, known as static implicit conditioning Meng et al. (2022), is insufficient to address the entire range of anomaly types and sizes. Therefore, we propose to compute the number of noising steps (noise amount) as a function of the input image and nominal training set, see Figure 2. This dynamic adjustment aids in precise segmentation of anomalies, which is often the weakest attribute of diffusion models in comparison with representation-based methods. To further abstract away from pixel-level details, we adopt a latent diffusion model and show that a latent representation along with the corresponding reconstruction provides state-of-the-art anomaly heatmaps while requiring less computing resources. Finally, our framework does not require noise to be added at inference time whereby a test image is directly denoised into a predicted reconstruction.

Our main contributions are as follows:

- We propose a dynamic conditioning mechanism where the maximum noise is computed using prior information about the anomaly provided by a KNN model of domain adapted features.
- We propose a domain adaptation mechanism that aims to learn the target domain as well as reconstruction errors.
- We propose to train a latent diffusion model for the task of anomaly detection to achieve precise anomaly heatmaps.
- We perform extensive evaluation and ablation studies on our approach and demonstrate state-of-the-art performance in segmentation of anomalies at all scales.

## 2  RELATED WORK

**Reconstruction Methods**   These methods hinge on the premise that trained models are unable to generate anomalies, resulting in large disparity between an anomalous input and its reconstruction. Autoencoders have been vastly explored Bergmann et al. (2019c); Gong et al. (2019), however, the reconstructions often include the anomalous region resulting in erroneous anomaly heatmaps. An improvement has been to combine (variational) Autoencoder Kingma & Welling (2014) with adversarial training, leveraging a discriminator, to spot anomalies Baur et al. (2019); Sabokrou et al.

(2018). However, these methods still suffer from significant reconstruction error. GANs have also been explored for anomaly detection. For instance, Schlegl et al. (2017) introduced a feature-wise and visual loss. In their approach, nearest latent representation of input images is iteratively sought. In contrast, Akcay et al. (2019) employed an encoder-decoder-encoder architecture, optimizing both image and latent representation reconstructions. A discriminator then compared features from the original and reconstructed images. Alternative techniques, as cited in Haselmann et al. (2018); Zavrtanik et al. (2021b); Ristea et al. (2022), approach the problem as an in-painting task whereby random patches from images are obscured, and neural networks learn to infer the missing data. DRAEM Zavrtanik et al. (2021a) used an end-to-end approach relying on synthetic data. Though reconstruction-based methods have had some success, they suffer from generated anomalies or artifacts within the reconstructions. Recent innovation have explored the potential of diffusion models in AD making use of an implicit conditioning proposed by SDEdit Meng et al. (2022). Works by Wyatt et al. (2022); Zhang et al. (2023); Mousakhan et al. (2023) have showcased success in achieving high quality anomaly heatmaps however, these approaches fail in the face of large sized defects. Our D3AD method is agnostic to anomaly size and is capable of detecting a wide range of anomalies with varying severity.

**Representation Methods** These methods gauge the discrepancy between the feature representation of test data and the learned representations of nominal data. This learned representation might either be a prototypical representation or the feature space mapping itself. PaDim Defard et al. (2021) employs a patch-wise extraction and concatenation of features from multiple CNN layers. An empirical sample mean and covariance matrix for each patch's feature vector is then computed. Anomalies are pinpointed based on the Mahalanobis distance between patches. Spade Cohen & Hoshen (2020) emphasizes this distance principle, computing the average distance of an image to its k-nearest neighbours pixel-wise and thresholding to discover anomalies. Patchcore Roth et al. (2022) is a synthesis of both PaDim and Spade, employing a patch strategy, with each patch being compared to a coreset of all other patches. The distance comparison mirrors Spade, focusing on the average distance to k-nearest neighbours within the coreset. Similarly CFA Lee et al. (2022) combines the patch based approach with metric learning. Another line of work utilises normalising flows Rudolph et al. (2020); Yu et al. (2021); Gudovskiy et al. (2022) to directly estimate the likelihood function whereby sample in the low-density regions can instantly be identified as anomalies. Nonetheless, none of these approaches generate an anomaly-free rendition of the input image. This capability is highly sought after in an industrial context, as it fosters trust and provides valuable insights into the model's decision-making process.

**Domain Adaption** Most prior approaches employ pretrained feature extractors to map raw images into a latent space. However, these feature extractors often lack adaptation to the target domain, resulting in artifacts for reconstruction-based methods and inaccuracies in representation-based comparisons. To address this, domain adaptation techniques have been explored. For instance, SimpleNet Liu et al. (2023) enhances a pretrained feature extractor with a domain adaptation layer and uses Gaussian noise to perturb features and training a discriminator to distinguish native from perturbed features. In contrast, RD4AD Deng & Li (2022) adopts an encoder-decoder structure, with the student network receiving the teacher's latent representation instead of the original image. RD++ Tien et al. (2023) extends this approach by incorporating additional projection layers to filter out anomalous information. Inspired by these successes, we implement a fine-tuning strategy for the pretrained feature extractors in order to leverage the demonstrated benefit.

## 3 BACKGROUND

We use a class of generative models called diffusion probabilistic models Sohl-Dickstein et al. (2015); Ho et al. (2020). In these, parameterized Markov chains with $T$ steps are used to gradually add noise to input data $x_0 \sim q(x_0)$ until all information is lost. The inspiration stems from principles of nonequilibrium thermodynamics Sohl-Dickstein et al. (2015). Neural networks are then parameterised to learn the unknown reverse process, in effect learning a denoising model. The forward process $q$ is defined as:

$$q(\boldsymbol{x}_t|\boldsymbol{x}_{t-1}) = \mathcal{N}(\boldsymbol{x}_t; \sqrt{1-\beta_t}\boldsymbol{x}_{t-1}, \beta_t\mathbf{I}) \tag{1}$$

$$q(\boldsymbol{x}_t|\boldsymbol{x}_0) = \mathcal{N}(\boldsymbol{x}_t; \sqrt{\bar{\alpha}_t}\boldsymbol{x}_0, (1-\bar{\alpha}_t)\mathbf{I}) \tag{2}$$

$$\boldsymbol{x}_t = \sqrt{\bar{\alpha}_t}\boldsymbol{x}_0 + \sqrt{1-\bar{\alpha}_t}\boldsymbol{\epsilon}, \quad \text{where} \quad \boldsymbol{\epsilon} \sim \mathcal{N}(0, \mathbf{I}) \tag{3}$$

Usually the $\beta_t$ are chosen as hyperparameters of the form $\beta_t \in (0,1)$ with a variance schedule $\beta_0 < \beta_1 < ... < \beta_T$ such that the signal of the input gets sequentially disturbed. For direct sampling the $\beta_t$ parameters are simplified to a compacter notation: $\alpha_t = 1 - \beta_t$ and $\bar{\alpha}_t = \prod_{s=1}^{t}\alpha_s$. Furthermore with large $T$ and small $\beta_t$, the distribution of $x_T$ approaches a standard normal which enables sampling from a normal distribution in the reverse process $p$ parameterized by $\theta$. This is defined as:

$$p_\theta(\boldsymbol{x}_{t-1}|\boldsymbol{x}_t) = \mathcal{N}(\boldsymbol{x}_{t-1}; \boldsymbol{\mu}_\theta(\boldsymbol{x}_t, t), \beta_t\mathbf{I}) \tag{4}$$

This corresponds to the DDPM Ho et al. (2020) formulation, where the variance is equivalent to the forward process while other works found better performance with learning the covariance matrix Nichol & Dhariwal (2021). DDPM is trained by predicting the initially added noise $\boldsymbol{\epsilon}$ which corresponds to predicting $\boldsymbol{\mu}_\theta$ and leads to the training objective:

$$L_{simple}(\theta) = \mathbb{E}_{t,\boldsymbol{x}_0,\boldsymbol{\epsilon}}[||\boldsymbol{\epsilon} - \boldsymbol{\epsilon}_\theta(\sqrt{\bar{\alpha}_t}\boldsymbol{x}_0 + \sqrt{1-\bar{\alpha}_t}\boldsymbol{\epsilon}, t)||_2^2] \tag{5}$$

The noising and denoising is performed in pixel space which is computationally expensive therefore Rombach et al. (2022) proposed to utilise latent spaces. An encoder $\mathcal{E}$ of a continuous or quantized VAE is used to project an image $\boldsymbol{x}_0$ into a lower dimension $\boldsymbol{z}_0 = \mathcal{E}(\boldsymbol{x}_0)$ while a decoder $\mathcal{D}$ aims to reconstruct this such that $\boldsymbol{x}_0 \simeq \hat{\boldsymbol{x}}_0 = \mathcal{D}(\boldsymbol{z}_0)$. The following objective function is used:

$$L_{simple-latent}(\theta) = \mathbb{E}_{t,\mathcal{E}(\boldsymbol{x}_0),\boldsymbol{\epsilon}}[||\boldsymbol{\epsilon} - \boldsymbol{\epsilon}_\theta(\sqrt{\bar{\alpha}_t}\boldsymbol{z}_0 + \sqrt{1-\bar{\alpha}_t}\boldsymbol{\epsilon}, t,)||_2^2] \tag{6}$$

A faster sampling approach is proposed by DDIM Song et al. (2022) where a non-Markovian formulation of the DDPM objective is employed allowing sampling steps to be omitted. This implies that a diffusion model trained according to objective Eq. 5 or Eq. 6 can be used to accelerate the sampling without the need for retraining. Their proposed sampling procedure is:

$$\boldsymbol{x}_{\tau_{i-1}} = \sqrt{\bar{\alpha}_{\tau_{i-1}}}\boldsymbol{f}_\theta^{(\tau)}(\boldsymbol{x}_\tau) + \sqrt{1-\bar{\alpha}_{\tau_{i-1}} - \sigma_{\tau_i}^2}\boldsymbol{\epsilon}_\theta(\boldsymbol{x}_{\tau_i}, \tau_i) + \sigma_{\tau_i}\boldsymbol{\epsilon}_{\tau_i} \tag{7}$$

Here $\tau_i, i \in [1, ..., S]$ acts as an index for subset $\{\boldsymbol{x}_{\tau_1}, ..., \boldsymbol{x}_{\tau_S}\}$ of length $S$ with $\tau$ as increasing sub-sequence of $[1, ..., T]$. Moreover, an estimation of $\boldsymbol{x}_0$ is obtained at every time step, denoted by $\boldsymbol{f}_\theta^{(t)}(\boldsymbol{x}_t) = \frac{\boldsymbol{x}_t - \sqrt{1-\bar{\alpha}_t}\boldsymbol{\epsilon}_\theta(\boldsymbol{x}_t, t)}{\sqrt{\bar{\alpha}_t}}$ which utilizes the error prediction $\boldsymbol{\epsilon}$ according to equation 3. DDIM further demonstrates varying levels of stochasticity within the model with also a fully deterministic version which corresponds to $\sigma_{\tau_i} = 0$ for all $\tau_i$.

Guidance and conditioning the sampling process of diffusion models has been recently explored and often requires training on the conditioning with either an extra classifier Dhariwal & Nichol (2021) or classifier-free guidance Ho & Salimans (2021). Recent work on AD with diffusion models Mousakhan et al. (2023) showed a guiding mechanism which does not require explicit conditional training. Guidance is achieved directly during inference by updating the predicted noise term using $\boldsymbol{x}_0$ or respectively $\boldsymbol{z}_0$ as:

$$\hat{\boldsymbol{\epsilon}}_t = \boldsymbol{\epsilon}_\theta(\boldsymbol{x}_t, t) - \eta\sqrt{1-\bar{\alpha}_t}(\tilde{\boldsymbol{x}}_t - \boldsymbol{x}_t) \quad \text{with} \quad \tilde{\boldsymbol{x}}_t = \sqrt{\bar{\alpha}_t}\boldsymbol{x}_0 + \sqrt{1-\bar{\alpha}_t}\boldsymbol{\epsilon}_\theta(\boldsymbol{x}_t, t) \tag{8}$$

where $\eta$ controls the temperature of guidance. This updated noise term can then be used in the DDIM sampling formulation 7 to result in the intended reconstruction $\hat{\boldsymbol{z}}_0$ and corresponding $\hat{\boldsymbol{x}}_0$.

## 4 METHOD

Diffusion models for AD learn the distribution of only nominal data such that they are unable to reconstruct anomalous regions leading to a large distance between input image $\boldsymbol{x}_0$ and it's reconstruction $\hat{\boldsymbol{x}}_0$. Previous approaches rely on implicit conditioning Meng et al. (2022), whereby the input is noised until a fixed time step $\hat{T} < T$ such that some input signal remains allowing for targeted reconstruction. We improve on this in two ways, first we discover that an noiseless and only scaled input $\boldsymbol{x}_{\hat{T}} = \boldsymbol{x}_0\sqrt{\bar{\alpha}_{\hat{T}}}$ is optimal for anomaly segmentation since it sufficiently reinforces the implicit conditioning applied on the model. Second we propose to choose forward time step $\hat{T}$ dynamically based on an initial estimate of the anomaly. Furthermore, we adopt the architecture of

unconditional latent diffusion model to abstract away from pixel-level representation which allows for improved reconstruction of large anomalies such as missing components in a resource efficient latent space. Our reconstruction and dynamic implicit conditioning frameworks are illustrated in Figure 3. Algorithm 1 describes the reconstruction process where we utilise the error correction (lines 6 and 7), proposed by DDAD Mousakhan et al. (2023), for guidance and the DDIM (Eq. 7) sampling procedure. Algorithm 2 details our dynamic conditioning mechanism for selecting optimum $\hat{T}$ for the forward process. Training the diffusion model is according to the objective function in Eq. 6 without modifications.

## 4.1 DYNAMIC IMPLICIT CONDITIONING

We introduce dynamic implicit conditioning (DIC) into the model's architecture. Specifically, we set a maximum implicit conditioning level denoted by $T_{max} \in \{1, ..., T\}$. This is selected such that the signal-to-noise ratio remains high. We then establish a quantization of the maximum steps into increments ranging up to $T_{max}$ with which we compute the dynamic implicit conditioning level $\hat{T}$ for each image according to an initial estimate of the anomaly.

**Bin construction** Our quantization is founded upon equidistant bins denoted as $b \in B$. These bins are determined from the average KNN distances of the training set's feature representations. Given that $\phi$ is a pretrained domain adapted feature extractor, and $\phi_j$ outputs the feature map of the $j^{\text{th}}$ layer block, for data point $x_0 \in \mathcal{X}_{Train}$, the features are extracted as $y_0 = \phi_j(x_0)$ with $y_0 \in Y_{Train}$. Utilizing $y_0$, a KNN-search is executed on the entire feature training set $Y_{Train}$ using the $\mathcal{L}1$-Norm. The K-nearest neighbors of $y_0$ are represented by the set $\{y_{s_1}, ..., y_{s_K}\}$. Subsequently, we compute the mean distance to these KNNs and denote it as $\bar{y}_0$. While this method is susceptible to outliers due to its reliance on the arithmetic mean, it is anticipated that anomalous data will be substantially more distinct than regular data. Thus, any outlier within the regular data would be beneficial as it would lead to a wider range for the bins. We compute the average distance for each sample in the training set. Furthermore using the computed average distances, we delineate $|B|$ evenly spaced bins.

**Dynamic Implicit Conditioning (DIC)** We denote DIC by function $g(x_0, \mathcal{X}_{Train}, T_{max})$ described in Algorithm 2. A visual representation of this mechanism is illustrated in Figure 3. During inference, for a new image $x_0$, we first utilise $\phi_j$ to extract features of $x_0$ and perform a KNN search on $Y_{train}$. The distances are averaged to compute $\bar{y}_0$ which is then placed into bin $b$ via a binary search function $\psi$ on all $b \in B$. The selected bin $b$ serves as an initial estimate of the severity of the anomaly in the input image compared to the nominal training data. The dynamic time step $\hat{T}$ is then simply computed as a fraction of $T_{max}$ based on the selected bin.

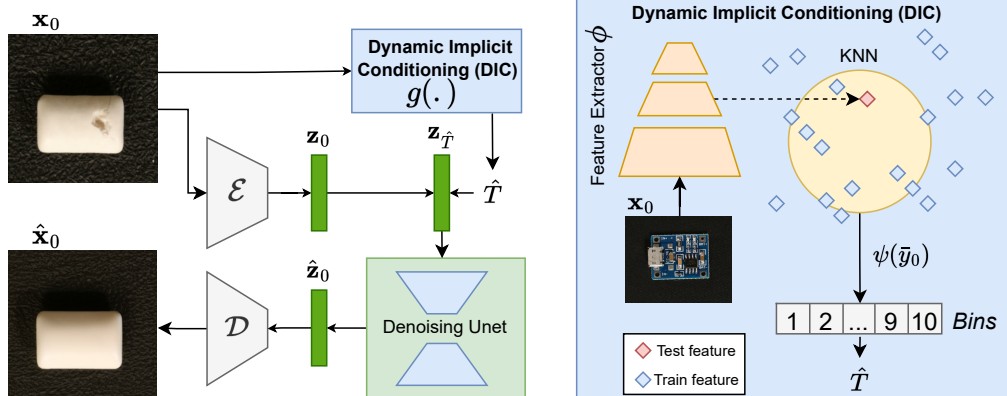

Figure 3: **Reconstruction Architecture**: An input $x_0$ is fed to the DIC to determine the level it must be perturbed $\hat{T}$. $x_0$ is also projected to a latent representation $z_0$. Denoising is performed in the latent space leading to a predicted latent $\hat{z}_0$ which is decoded into a reconstruction $\hat{x}_0$. **DIC**: The average distance of extracted features of a test image to the K nearest neighbours from the training set is quantized, using equally sized predefined bins, to then determine the dynamic noising step $\hat{T}$.

**Algorithm 1** Dynamic Reconstruction

1: **input** $x_0$
2: $\hat{T} = g(x_0, \mathcal{X}_{Train}, T_{max})$
3: $z_0 = \mathcal{E}(x_0)$
4: $z_{\hat{T}} = z_0\sqrt{\bar{\alpha}_{\hat{T}}}$   # no noise
5: **for** $t = \hat{T}, ..., 1$ **do**
6:     $\tilde{z}_t = \sqrt{\bar{\alpha}_t}z_0 + \sqrt{1-\bar{\alpha}_t}\epsilon_\theta(z_t, t)$
7:     $\hat{\epsilon}_t = \epsilon_\theta(z_t, t) - \eta\sqrt{1-\bar{\alpha}_t}(\tilde{z}_t - z_t)$
8:     $\hat{z}_{t-1} = \sqrt{\bar{\alpha}_{t-1}}z_{\theta,0} + \sqrt{1-\bar{\alpha}_{t-1}}\hat{\epsilon}_t$
9: **end for**
10: $\hat{x}_0 = \mathcal{D}(\hat{z}_0)$
11: **return** $\hat{x}_0, \hat{z}_0$

**Algorithm 2** Dynamic Implicit Conditioning $g$

1: **input** $x_0$
2: **input** $T_{max}$
3: $Y_{Train} = \phi_j(\mathcal{X}_{Train})$
4: $y_0 = \phi_j(x_0)$
5: $\{y_{s_1}, ..., y_{s_K}\} = \text{KNN}(y_0, Y_{train}, K)$
6: $\bar{y}_0 = \frac{1}{K}\sum_{j=1}^{K} ||y_0 - y_{s_j}||$
7: $b = \psi(\bar{y}_0)$   # binary search
8: $\hat{T} = \left\lfloor \frac{b}{|B|}T_{max} \right\rfloor$
9: **return** $\hat{T}$

## 4.2 ANOMALY SCORING AND MAP CONSTRUCTION

We adopt the convention of comparing the input image with its reconstruction to generate the final anomaly map as illustrated in Figure 4. We compare the latent representation $z_0$ with its reconstruction $\hat{z}_0$ to construct a latent anomaly map $l_{map}$. Similarly, we compare the features of the input image $x_0$ against its reconstruction $\hat{x}_0$ to construct a feature anomaly map $f_{map}$. A weighted combination generates the final anomaly map $A_{map}$.

The feature anomaly map $f_{map}$ is determined by first computing the features of an input image $x_0$ and its reconstruction $\hat{x}_0$ using a pretrained and domain adapted feature extractor $\phi$ (section 4.3). A cosine distance between the extracted feature blocks at $\mathbb{J} \subseteq \{1, ..., J\}$ layers of a ResNet-34 yields the feature anomaly map. Given that feature blocks at different layers may present divergent dimensionalities, these are upsampled to achieve uniformity. The feature anomaly map $f_{map}$ is articulated as $f_{map}(x_0, \hat{x}_0) = \sum_{j \in \mathbb{J}} (cos_d(\phi_j(x_0), \phi_j(\hat{x}_0)))$.

Since our approach relies on learning a denoising diffusion model on the latent representation, we further compute distances between the input image latent representation $z_0$ and its reconstructed counterpart $\hat{z}_0$. Utilizing the $\mathcal{L}1$-Norm for each pixel, a latent anomaly map is deduced as $l_{map}(z_0, \hat{z}_0) = ||z_0 - \hat{z}_0||_1$.

The final anomaly map $A_{map}$ is simply a linear combination of the normalized feature-based distance and the latent pixel-wise distance as follows:

$$A_{map} = \lambda * l_{map}(z_0, \hat{z}_0) + (1-\lambda) * f_{map}(x_0, \hat{x}_0) \tag{9}$$

Subsequently, an established threshold facilitates the categorization of every pixel and image, marking them as either anomalous or typical. The global image anomaly score is selected as the maximum pixel-level anomaly score within the entire image.

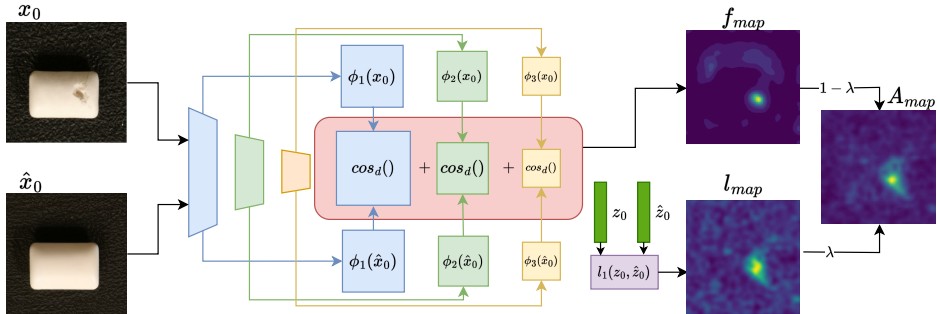

Figure 4: Overview of the Anomaly Map construction. Feature heatmap ($f_{map}$) are computed as cosine distances of the features of the input $x_0$ and its reconstruction $\hat{x}_0$ whereas latent heatmap ($l_{map}$) is calculated using an $\mathcal{L}1$ distance between the corresponding latent representations of $x_0$ and $\hat{x}_0$. These combine linearly to form the final anomaly heatmap ($A_{map}$).

## 4.3 DOMAIN ADAPTATION

We leverage domain-adapted features for both the dynamic implicit conditioning and the construction of the feature anomaly map $f_{map}$. Our objective is to grasp the intricacies associated with the target domain. With the use of variational autoencoders (VAEs) having pretrained encoders and decoder introduces artifacts and reconstruction inaccuracies. These are incorrectly flagged as anomalous regions during comparison. To address this, we introduce a loss function to fine-tune the feature extractor $\phi$ by further training for $\gamma$ epochs. This function is designed to minimize the feature distance between the input image $\boldsymbol{x}_0$ and its reconstruction $\hat{\boldsymbol{x}}_0$ as follows where GAP refers to global average pooling:

$$L_{DA}(\boldsymbol{x}_0, \hat{\boldsymbol{x}}_0) = \sum_{j=1}^{J} \text{GAP} \left( 1 - \frac{\phi_j(\boldsymbol{x}_0)^T \phi_j(\hat{\boldsymbol{x}}_0)}{||\phi_j(\boldsymbol{x}_0)|| \, ||\phi_j(\hat{\boldsymbol{x}}_0)||} \right). \tag{10}$$

## 5 EXPERIMENTS

**Datasets**  We employ two widely used benchmarking datasets to evaluate the veracity of our appraoch, namely **VisA** Zou et al. (2022) and **BTAD** Mishra et al. (2021) dataset. VisA dataset presents a collection of 10,821 high-resolution RGB images, segregated into 9,621 regular and 1,200 anomalous instances. Comprehensive annotations are available in the form of both image and pixel-level labels. The dataset comprises of 12 different classes with a large variety of scale and type of anomalies. BTAD dataset comprises of RGB images showcasing three unique industrial products. There are 2540 images in total where each anomalous image is paired with a pixel-level ground truth mask.

**Evaluation Metrics**  We evaluate our approach using standard metrics for anomaly detection, namely pixel-wise AUROC (P-AUROC), image-wise AUROC (I-AUROC) and the PRO metric. P-AUROC is ascertained by setting a threshold on the anomaly score of individual pixels. A critical caveat of P-AUROC is its potential for overestimation, primarily because a majority of pixels are typically normal. Such skewed distribution occasionally renders a misleadingly optimistic performance portrayal. Addressing this limitation, the PRO metric Bergmann et al. (2019a) levels the playing field by ensuring equal weighting for both minuscule and pronounced anomalies. This balance is achieved by averaging the true positive rate over regions defined by the ground truth, thereby offering a more discerning evaluative metric making it our primary choice for evaluation. The image-wise AUROC (I-AUROC) is employed to present an evaluation of image-based anomaly detection, where precise segmentation of the anomaly is unimportant.

**Implementation Details**  We employ an unconditional Unet from Rombach et al. (2022) with an 8x downsampling within our diffusion model. For KNN, we set $K = 20$ with $\mathcal{L}1$ distance. Both dynamic conditioning and anomaly map construction utilize a ResNet-34 pretrained on ImageNet and fine-tuned. Domain adaptation is performed for up to 3 epochs using identical Unet settings. $T_{max}$ is set at 80 for VisA and remains unchanged for BTAD. We chose $|B| = 10$ which leads to a percentage-quantization mapping of increments of 10% steps of $T_{max}$. However, we set the minimum bin to 2, ensuring that we don't rely solely on prior information. Lastly, the DDIM formulation with 10 steps is adopted for sampling, with the DIC step rounded to the nearest multiple of 10. All experiments were carried out on one Nvidia RTX 8000. Further implementational details are present in Appendix A.1.

**Anomaly Detection Results**  We conduct comprehensive experiments on the VisA dataset to evaluate the capability of our proposed method in detecting and segmenting anomalies. Table 1 details the performance of our method. Notably, D3AD excels in 8 of the 12 classes in segmentation accuracy as evident from PRO values, and in 3 of 12 classes for I-AUROC whilst achieving comparable performance in remaining classes. The aggregate performance across all classes yields an I-AUROC of 96.0%, paralleling the performance of the state-of-the-art method, RD4AD. Whereas there is a clear superiority of our method in segmentation achieving an average of 94.1%, outperforming the contemporary state-of-the-art by 2.7% points.

In an evaluation alongside other diffusion-based models, as documented in Table 2, D3AD achieves superior anomaly localisation performance on the VisA benchmark. When assessed using PRO and

P-AUROC, 3DAD demonstrates an enhancement, achieving results higher by at least $0.9\%$ points for both metrics compared to previous diffusion state-of-the-art approaches. Figure 1 offers a teaser of D3AD's qualitative performance, with a comprehensive evaluation provided in appendix A.2. Significantly, the method excels in precise segmentation and effectively handling large anomalies.

Further results from the BTAD benchmark are consolidated in Table 3. Here, D3AD exhibits competitive performance in terms of I-AUROC. More prominently, and following previous trend, segmentation evaluated using PRO highlight our method achieving unparalleled results, surpassing the closest competitors by a margin of $5.9$ percentage points.

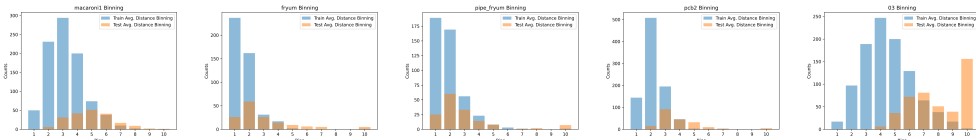

Figure 5: Histogram of the binning values for the training set in blue and test set in orange, showing a distribution shift to larger values for the test set. Displayed are categories from VisA and BTAD.

Table 1: Anomaly classification and localization performance (I-AUROC, PRO) of various methods on VisA benchmark. The best results are highlighted in bold.

| | Representation-based | | | | Reconstruction-based | |
|---|---|---|---|---|---|---|
| **Method** | SPADE | PaDiM | RD4AD | PatchCore | DRAEM | D3AD (Ours) |
| **Candle** | (91.0,93.2) | (91.6,**95.7**) | (92.2,92.2) | (**98.6**,94.0) | (91.8,93.7) | (95.6,92.7) |
| **Capsules** | (61.4,36.1) | (70.7,76.9) | (**90.1**,56.9) | (81.6,85.5) | (74.7,84.5) | (88.5,**95.7**) |
| **Cashew** | (97.8,57.4) | (93.0,87.9) | (**99.6**,79.0) | (97.3,**94.5**) | (95.1,51.8) | (94.2,89.4) |
| **Chewing gum** | (85.8,93.9) | (98.8,83.5) | (**99.7**,92.5) | (99.1,84.6) | (94.8,60.4) | (**99.7**,**94.1**) |
| **Fryum** | (88.6,91.3) | (88.6,80.2) | (96.6,81.0) | (96.2,85.3) | (**97.4**,**93.1**) | (96.5,91.7) |
| **Macaroni1** | (95.2,61.3 ) | (87.0,92.1) | (**98.4**,71.3) | (97.5,95.4) | (97.2,96.7) | (94.3,**99.3**) |
| **Macaroni2** | (87.9,63.4) | (70.5,75.4) | (**97.6**,68.0) | (78.1,94.4) | (85.0,92.6) | (92.5,**98.3**) |
| **PCB1** | (72.1,38.4) | (94.7,91.3) | (97.6,43.2) | (**98.5**,94.3) | (47.6,24.8) | (97.7,**96.4**) |
| **PCB2** | (50.7,42.2) | (88.5,88.7) | (91.1,46.4) | (97.3,89.2) | (89.8,49.4) | (**98.3**,**94.0**) |
| **PCB3** | (90.5,80.3) | (91.0,84.9) | (95.5,80.3) | (**97.9**,90.9) | (92.0,89.7) | (97.4,**94.2**) |
| **PCB4** | (83.1,71.6) | (97.5,81.6) | (96.5,72.2) | (99.6,**90.1**) | (98.6,64.3) | (**99.8**,86.4) |
| **Pipe fryum** | (81.1,61.7) | (97.0,92.5) | (97.0,68.3) | (99.8,95.7) | (**100**,75.9) | (96.9,**97.2**) |
| **Average** | (82.1,65.9) | (89.1,85.9) | (**96.0**,70.9) | (95.1,91.2) | (88.7,73.1) | (**96.0**,**94.1**) |

**Ablation Studies** To understand the significance of each component in our D3AD model, we executed an ablation study using the VisA dataset to evaluate our proposed dynamic implicit conditioning mechanism, domain adapted feature extractor and input scaling without noising method.

Table 4 delves into the efficacy of our dynamic implicit conditioning (DIC). The DIC was compared against each quartile of the selected $T_{max}$, ranging from 25% to 100% of 80. The DIC consistently registered superior I-AUROC and P-AUROC scores, surpassing the second-best 80-step static model by margins of 0.6 and 1.2 percentage points, respectively. While PRO scores remained fairly consistent across different maximum step choices, the 20-step model slightly outperformed others with a score of 94.3, a slender 0.2 percentage points above the DIC. Given that PRO evaluates anomalies

Table 2: Detection and segmentation performance of diffusion based methods (AnoDDPM Wyatt et al. (2022), DiffusionAD Zhang et al. (2023), DDAD Mousakhan et al. (2023)) on VisA.

| Method | AnoDDPM | DiffusionAD | DDAD | D3AD (Ours) |
|---|---|---|---|---|
| I-AUROC | 78.2 | 97.8 | **99.3** | 96.0 |
| P-AUROC | - | - | 97.0 | **97.9** |
| PRO | 60.5 | 93.2 | 92.0 | **94.1** |

Table 3: Anomaly classification and localization performance (I-AUROC, PRO) of various methods on BTAD benchmark. The best results are highlighted in bold.

| | Representation-based | | | | | Reconstruction-based |
|---|---|---|---|---|---|---|
| **Method** | FastFlow | CFA | PatchCore | RD4AD | RD++ | D3AD (Ours) |
| **Class 01** | (**99.4**,71.7) | (98.1,72.0) | (96.7,64.9) | (96.3,75.3) | (96.8,73.2) | (98.9,**80.0**) |
| **Class 02** | (82.4,63.1) | (85.5,53.2) | (81.4,47.3) | (86.6,68.2) | (**90.1**,71.3) | (87.0,**71.7**) |
| **Class 03** | (91.1,79.5) | (99.0,94.1) | (**100.0**,67.7) | (**100.0**,87.8) | (**100.0**,87.4) | (99.7,**97.8**) |
| **Average** | (91.0,71.4) | (94.2,73.1) | (92.7,60.0) | (94.3,77.1) | (**95.6**,77.3) | (95.2,**83.2**) |

uniformly across all scales, and P-AUROC is more sensitive to large-scale anomalies, our observations suggest that the DIC adeptly identifies large anomalies, without compromising its efficiency across varying scales. The distribution of the initial signal is depicted in Figure 5 while Figure 6 shows the qualitative effect of DIC. It is apparent that a dynamically computed time step (DIC Mask) provided the most similar anomaly mask prediction to the ground truth (GT) mask, in comparison to fixed time step masks shown from 100% - 25% of $T$.

Table 5 illustrates the effects of the domain adaptation in the feature extractor and introducing a scaled, yet noiseless, input. Using a model without domain-adapted feature extraction and conventional noised input as the baseline, we observe notable improvements with the integration of each component. Particularly, the modified implicit conditioning, indicated as "downscaling (DS)" in the table, emerges as the most impactful modification. A detailed qualitative visualisation is shown in appendix Figures 10 to 13 whereas a quantitative study of this effect is present in Figure 14.

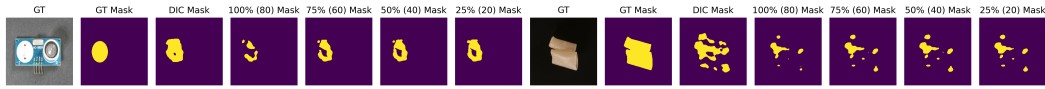

Figure 6: Overview of prediction masks for different levels of maximum static noise levels and the DIC. DIC tends to segment large anomalies more faithfully

Table 4: Impact of Dynamic Implicit Conditioning (DIC)

| Max. Step | Performance | | |
|---|---|---|---|
| | I-AUROC ↑ | PRO ↑ | P-AUROC ↑ |
| 25%(20) | 95.2 | **94.3** | 96.7 |
| 50%(40) | 94.7 | 94.1 | 96.6 |
| 75%(60) | 95.0 | 94.2 | 96.7 |
| 100%(80) | 95.4 | 94.0 | 96.7 |
| DIC($g(.)$) | **96.0** | 94.1 | **97.9** |

Table 5: Impact of Downscaling (DS) and Domain Adaptation (DA)

| Ablation | | Performance | | |
|---|---|---|---|---|
| DS | DA | I-AUROC ↑ | PRO ↑ | P-AUROC ↑ |
| - | - | 89.2 | 82.0 | 92.3 |
| ✓ | - | 95.4 | 92.0 | 96.9 |
| - | ✓ | 90.8 | 83.8 | 93.2 |
| ✓ | ✓ | **96.0** | **94.1** | **97.9** |

## 6 CONCLUSION

We propose to rethink the convention, of diffusion models for the unsupervised anomaly detection task, of noising all samples to the same time step and instead use prior information to dynamically adjust such implicit conditioning. Moreover we show that initial noising is counter productive and that a domain adapted feature extractor provides additional information for detection and localization. We introduced D3AD that combined all the proposed steps into an architecture which achieves state-of-the-art performance on the VisA benchmark with $96\%$ I-AUROC and $94.1\%$ PRO. Furthermore we showed that the segmentation performance measured by P-AUROC and PRO exceeds all previous suggested diffusion based models for unsupervised anomaly detection on VisA. A limitation of the framework is slower inference speed, which can potentially be addressed through innovations like precomputed features and more efficient approximations for anomaly severity, these are reserved for future work.

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

# A APPENDIX

## A.1 IMPLEMENTATION DETAILS

Training the UNET is conducted for 300 epochs using the AdamW optimizer Loshchilov & Hutter (2019). We set a learning rate of 0.0001 and weight decay to 0.01. The noise schedule is from 0.0015 to 0.0195 and we set $T = 1000$. For the ResNet-34, we set the dynamic conditioning feature blocks $\mathbb{J}$ to 2 whereas for anomaly map computation, features are extracted from blocks 2 and 3. The guidance temperature is either 0 (indicating no guidance) or within the range 7-10. We set the number of epochs $\gamma$ for fine-tuning the feature extractor in the range 0 to 3. The weighting parameter $\lambda$ for the anomaly maps is set to $0.85$ and the final anomaly map is smoothed with a Gaussian filter with $\sigma = 4$. The pretrained VAE from Rombach et al. (2022) is used without further training.

## A.2 ADDITIONAL DETAILS AND ABLATIONS

### A.2.1 NOISELESS RECONSTRUCTION

We studied the influence of the 'noiseless' and only scaled input on performance of the VisA benchmark. In figure 14, we provide different fractions of noise influence and the corresponding metrics. We tested fractions $\omega \in \{0, 0.1, 0.2, ..., 1\}$ of the noise as follows:

$$\boldsymbol{x}_t = \sqrt{\bar{\alpha}_t}\boldsymbol{x}_0 + \omega\sqrt{1 - \bar{\alpha}_t}\boldsymbol{\epsilon} \quad \text{where} \quad \boldsymbol{\epsilon} \sim \mathcal{N}(0, \mathbf{I}) \tag{11}$$

We perceived best performance with our proposed noiseless scaling ($\omega = 0$) with a declining performance as $\omega$ increases. In addition, we conducted a qualitative analysis to compare the visual impact of image-level perturbations in the forward diffusion process (as outlined in Equation 1; refer to Figures 11 and 12). Our tests extend up to the 400th time step, revealing that introducing noise degrades the visual quality of the signal rapidly. Furthermore, Figure 13 illustrates the effect of noiseless scaling over an extended period, up to the 800th time step. To provide a more transparent comparison, we executed these disturbance analyses in the pixel space rather than in the latent space. Figure 15 shows the anomaly map construction and reconstruction with varying perturbance levels for noiseless scaling versus noising. The Figure shows similar segmentation performance with slightly less artifacts in the anomaly map created by noiseless scaling, while for high perturbance levels (T=240,320) the noising paradigm is prone to hallucinations in the reconstruction as highlighted by a red circle.

### A.2.2 ADDITIONAL QUANTITATIVE ANALYSIS

To showcase the generalization capabilities of D3AD, we tested it on the MVTec Bergmann et al. (2019b) dataset, see Table 6. While representation-based methods perform decently, we beat the current reconstruction-based state-of-the-art model DRAEM Zavrtanik et al. (2021a) on the segmentation metrics P-AUROC and PRO by $0.1$ and $0.3$ percentage points respectively, however we trail in detection results I-AUROC by $0.8$ percentage points. Table 7 shows the localization performance measured by P-AUROC on the VisA benchmark.

### A.2.3 ADDITIONAL QUALITATIVE ANALYSIS

In Figure 8, we present a side-by-side comparison of our method's reconstruction capabilities against those of DRAEM Zavrtanik et al. (2021a). This comparison underscores a notable improvement in reconstruction quality achieved by our approach. Moreover, Figure 10 provides additional instances of anomaly segmentation, further illustrating our method's efficacy. Notably, Figure 7 encompasses both reconstruction and segmentation outcomes. The remarkable segmentation results are attributed to our method's robust reconstruction abilities and the utilization of domain-adapted feature signals. Our method's strength in reconstruction is bolstered by initially estimating the anomaly size, which allows for the effective scaling of large anomalies, as illustrated in rows 3-5 of Figure 7. Additionally, our approach demonstrates impeccable reconstruction of smaller defects, as shown in rows 1, 2, and 6-9, thanks to the selection of appropriate scaling levels. This aspect is further corroborated by Table 4 in the main document. The implementation of a noiseless, scaled latents further enhances these effects, as detailed in Figure 14 and discussed in Appendix section A.2.1. Furthermore, the domain-adapted feature extractor effectively learns the subtleties of the target domain, efficiently

filtering out any artifacts that may arise during the reconstruction process. Lastly, Figure 9 visually contrasts the segmentation efficiency between our method, DRAEM (Zavrtanik et al. (2021a)), DDAD (Mousakhan et al. (2023)), and D3AD.

### A.2.4 COMPUTATIONAL ANALYSIS

Lastly we present an evaluation on inference time and the frames per second (FPS) rate, as detailed in Table 8. We compare to various representation and reconstruction-based methods and achieve competitive performance. All experiments were carried out on one Nvidia Quadro 8000 graphics card, with a set batch size of 30. The evaluation for the baseline methods got performed with the Anomalib package Akcay et al. (2022).

Table 6: A detailed comparison of Anomaly Classification and localisation performance of various methods on MVTec benchmark Bergmann et al. (2019a) in the format of (I-AUROC, P-AUROC, PRO). Best results are highlighted in bold.

| | Representation-based | | | Reconstruction-based | | | |
|---|---|---|---|---|---|---|---|
| **Method** | PatchCore | SimpleNet | RD++ | GANomaly | SkipGANomaly | DRAEM | D3AD (Ours) |
| **Carpet** | (98.7,99.0,96.6) | (99.7,98.2,-) | (**100**,**99.2**,**97.7**) | (20.3,-) | (70.9,-) | (97.0,95.5,92.9) | (94.2,97.6,95.1) |
| **Grid** | (98.2,98.7,96.0) | (99.7,98.8,-) | (**100**,99.3,97.7) | ( 40.4,-) | (47.7,-) | (99.9,**99.7**,**98.4**) | (**100**,99.2,96.9) |
| **Leather** | (**100**,99.3,98.9) | (**100**,99.2,-) | (**100**,**99.4**,**99.2**) | ( 41.3,-) | (60.9,-) | (**100**,98.6,98.0) | (98.5,**99.4**,98.1) |
| **Tile** | (98.7,95.4,87.3) | (**99.8**,97.0,-) | (99.7,96.6,92.4) | (40.8 ,-) | (29.9,-) | (99.6,**99.2**,**98.9**) | (95.5,94.7,93.6) |
| **Wood** | (99.2,95.0,89.4) | (**100**,94.5,-) | (99.3,95.8,93.3,) | ( 74.4,-) | (19.9,-) | (99.1,**96.4**,**94.6**) | (99.7,95.9,91.0) |
| **Bottle** | (**100**,98.6,96.2) | (**100**,98.0,-) | (**100**,98.8,97.0) | (25.1,-) | (85.2,-) | (99.2,**99.1**,**97.2**) | (**100**,98.6,96.0) |
| **Cable** | (99.5,**98.4**,92.5) | (**99.9**,97.6,-) | (99.2,**98.4**,**93.9**) | ( 45.7,-) | (54.4,-) | (91.8,94.7,76.0) | (97.8,93.3,87.3) |
| **Capsule** | (98.1,98.8,95.5) | (97.7,**98.9**,-) | (**99.0**,98.8,96.4) | ( 68.2,-) | (54.3,-) | (98.5,94.3,91.7) | (96.6,97.9,90.7) |
| **Hazelnut** | (**100**,98.7,93.8) | (**100**,97.9,-) | (**100**,99.2,96.3) | ( 53.7,-) | (24.5,-) | (**100**,**99.7**,**98.1**) | (98.0,98.8,91.8) |
| **Metal nut** | (**100**,98.4,91.4) | (**100**,98.8,-) | (**100**,98.1,93.0) | (27.0 ,-) | (81.4,-) | (98.7,**99.5**,**94.1**) | (98.9,96.1,89.7) |
| **Pill** | (96.6,97.4,93.2) | (99.0,**98.6**,-) | (98.4,98.3,**97.0**) | ( 47.2,-) | (67.1,-) | (98.9,97.6,88.9) | (**99.2**,98.2,96.2) |
| **Screw** | (98.1,99.4,97.9) | (98.2,99.3,-) | (**98.9**,**99.7**,**98.6**) | ( 23.1,-) | (87.9,-) | (93.9,97.6,98.2) | (83.9,99.0,95.5) |
| **Toothbrush** | (**100**,98.7,91.5) | (99.7,98.5,-) | (**100**,**99.1**,94.2) | (37.2 ,-) | (58.6,-) | (**100**,98.1,90.3) | (**100**,99.0,**94.6**) |
| **Transistor** | (**100**,96.3,83.7) | (**100**,**97.6**,-) | (98.5,94.3,81.8) | ( 44.0,-) | (84.5,-) | (93.1,90.9,81.6) | (96.8,95.6,**86.9**) |
| **Zipper** | (99.4,98.8,**97.1**) | (99.9,**98.9**,-) | (98.6, 98.8,96.3) | (43.4 ,-) | (76.1,-) | (**100**,98.8,96.3) | (98.2,98.3,95.3) |
| **Average** | (99.1,98.1,93.4) | (**99.6**,98.1,-) | (99.4,**98.3**,**95.0**) | (42.1 ,-) | (60.2,-) | (98.0,97.3,93.0) | (97.2,97.4,93.3) |

Table 7: Localization performance (P-AUROC) of various methods on VisA benchmark. The best results are highlighted in bold.

| Method | SPADE | PaDiM | RD4AD | PatchCore | DRAEM | D3AD (Ours) |
|---|---|---|---|---|---|---|
| P-AUROC | 85.6 | 98.1 | 96.5 | **98.8** | 93.5 | 97.9 |

Table 8: Inference time for one image in seconds and frames-per-second (FPS) of selected models on VisA benchmark.

| | Representation-based | | Reconstruction-based | |
|---|---|---|---|---|
| **Method** | RD4AD | PatchCore | DRAEM | D3AD (Ours) |
| **FPS** | (4.8) | (4.8) | (4.3) | (2.9) |
| **Inference Time** | (0.21) | (0.21) | (0.23) | (0.34) |

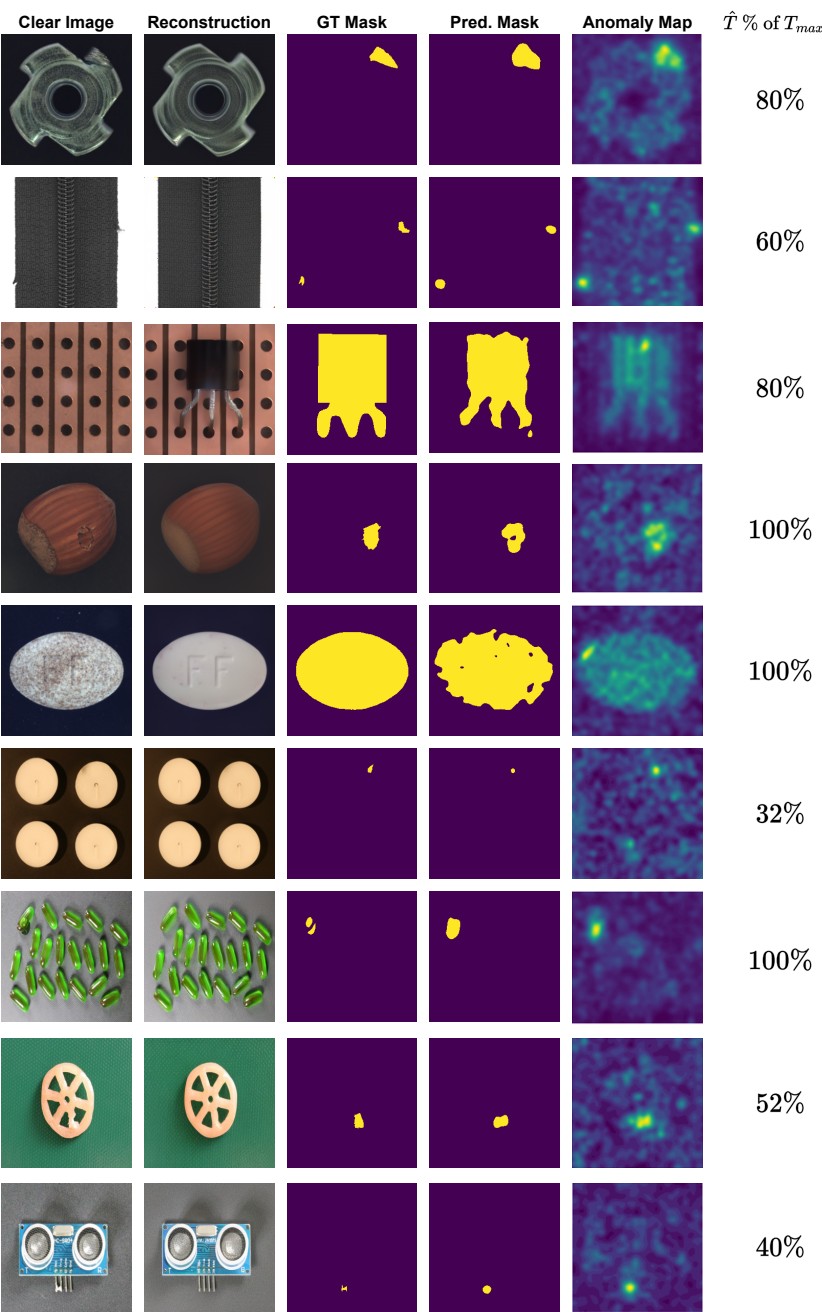

Figure 7: Reconstruction and segmentation performance of D3AD of various categories of the VisA and MVTec benchmark.

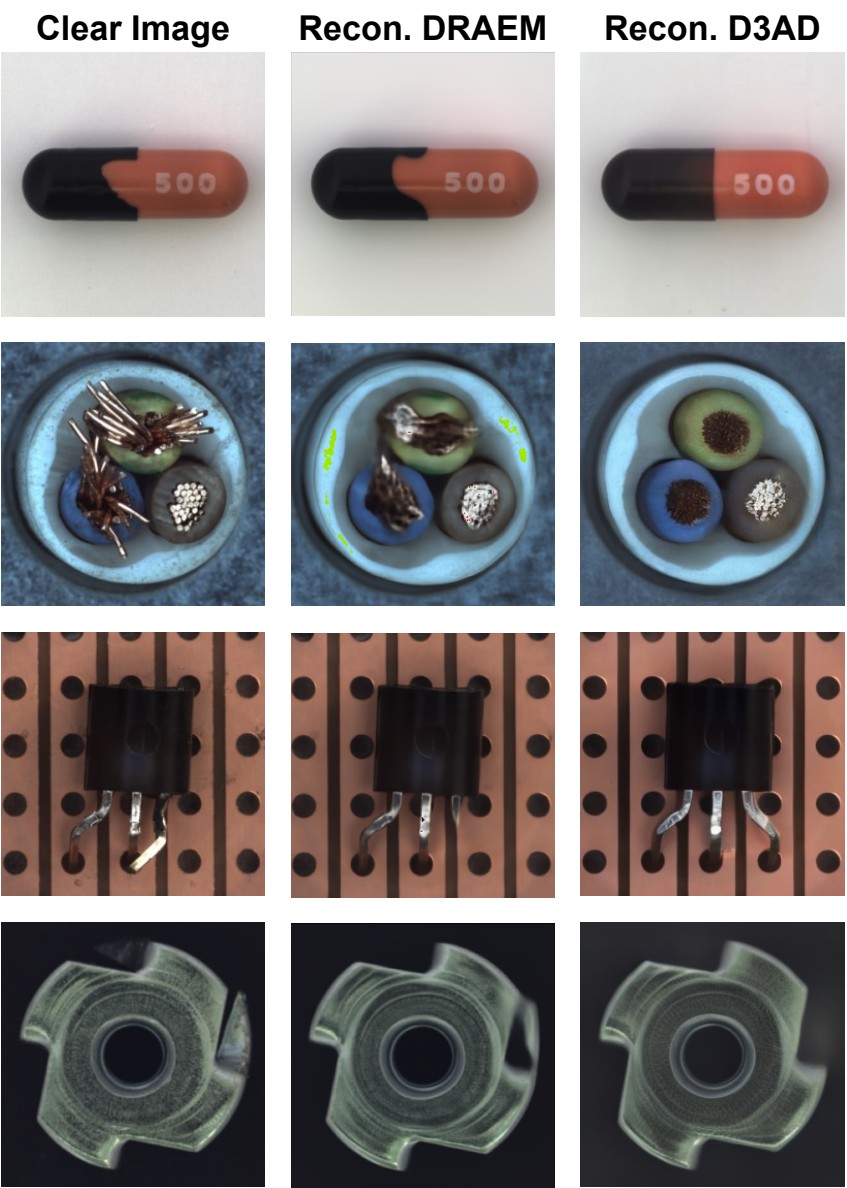

Figure 8: Reconstruction comparison with DRAEM Zavrtanik et al. (2021a) on various MVTec categories.

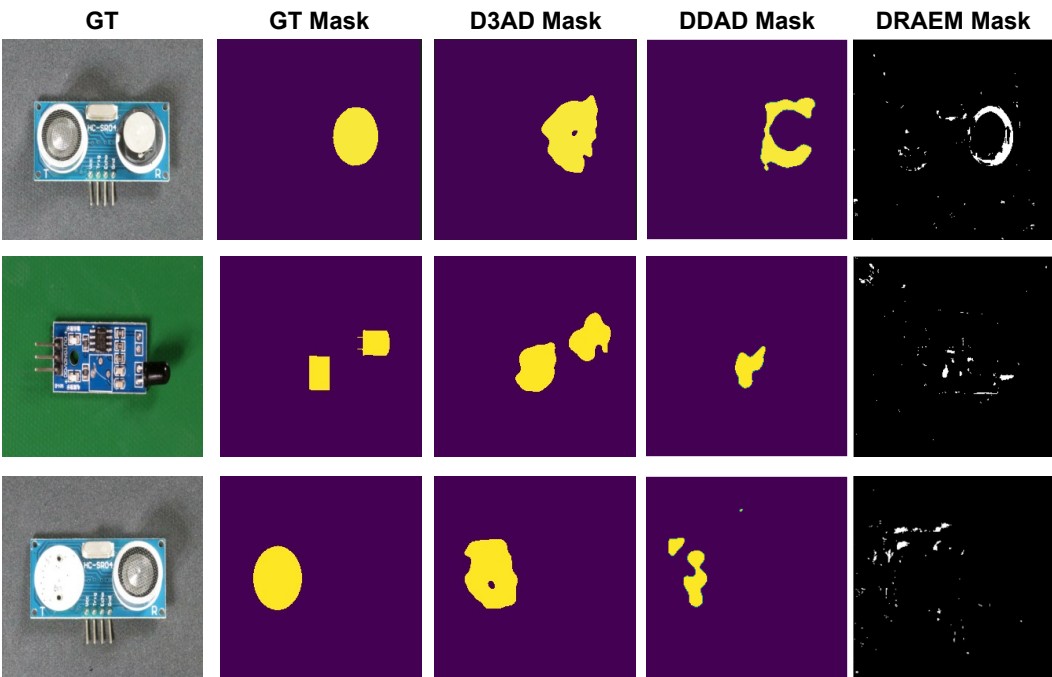

Figure 9: Mask comparison with DRAEM Zavrtanik et al. (2021a) and DDAD Mousakhan et al. (2023). Diffusion models and GT are depicted with yellow masks. D3AD shows improved segmentation abilities to previous approaches concerning large anomalies.

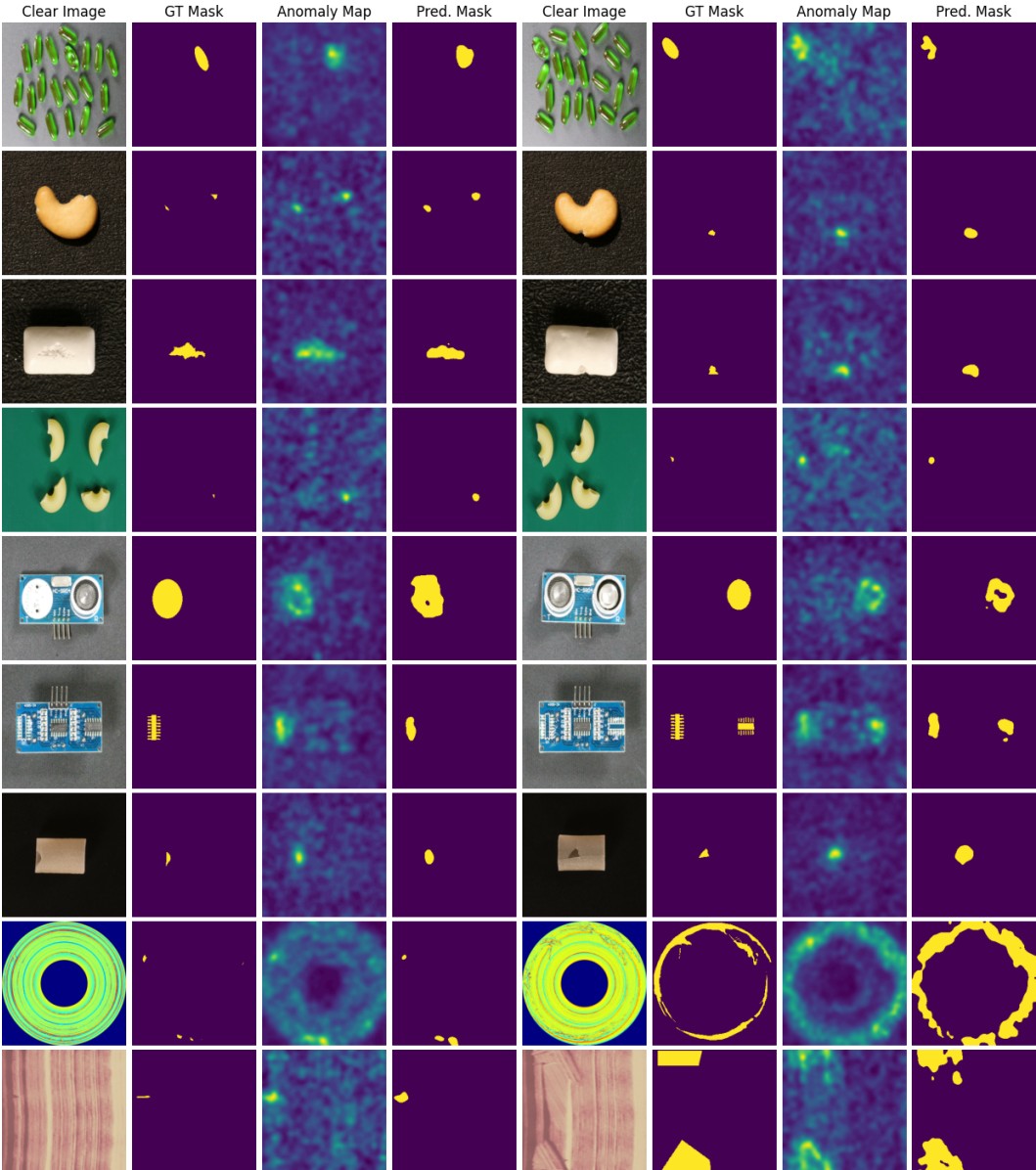

Figure 10: Additional examples from anomalies across all scales from the VisA and BTAD benchmark.

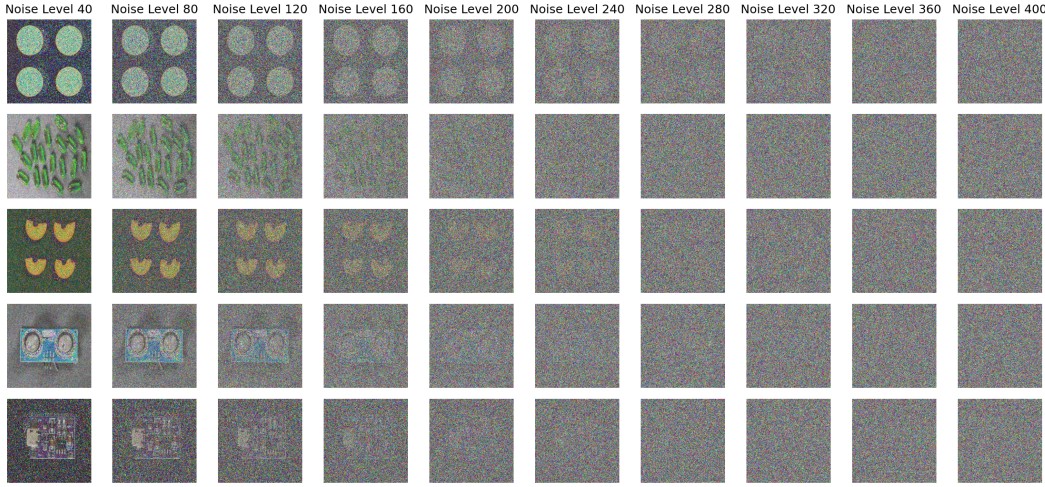

Figure 11: Visualization of the forward diffusion process in pixel space on various categories of the VisA benchmark.

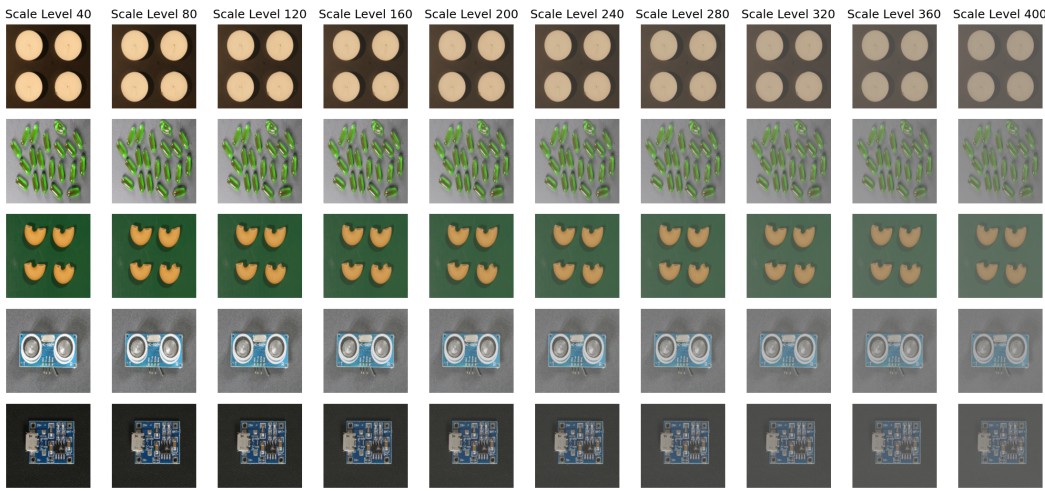

Figure 12: Visualization of the noiseless-forward scaling process in pixel space on various categories of the VisA benchmark.

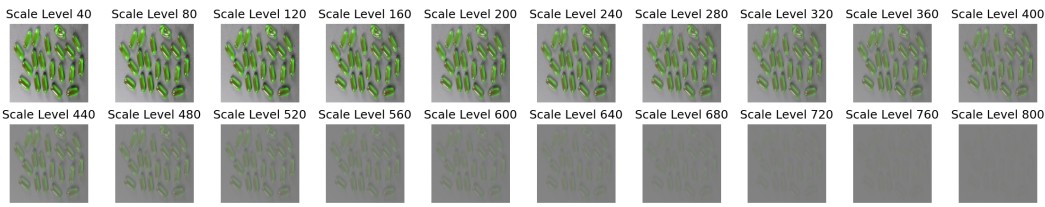

Figure 13: Visualization of the noiseless-forward scaling process in pixel space up to the time step $t = 800$ on the capsules category of the VisA benchmark.

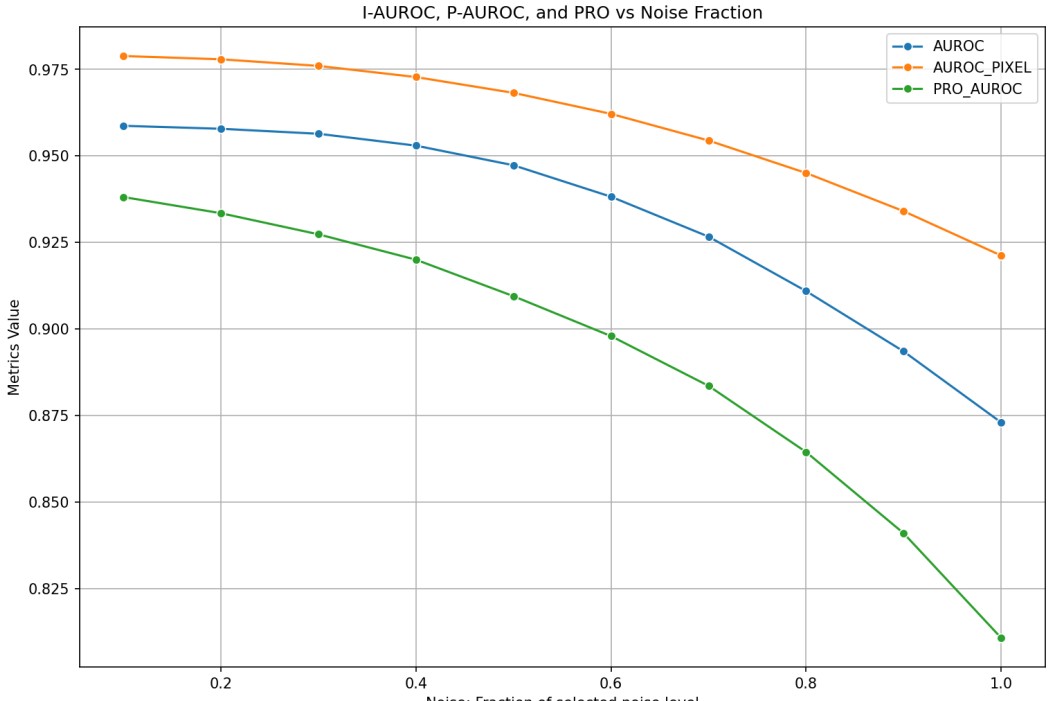

Figure 14: Impact of adding a fraction of the total noise on the VisA benchmark. Showcasing a decline in performance with increasing fraction of the noise.

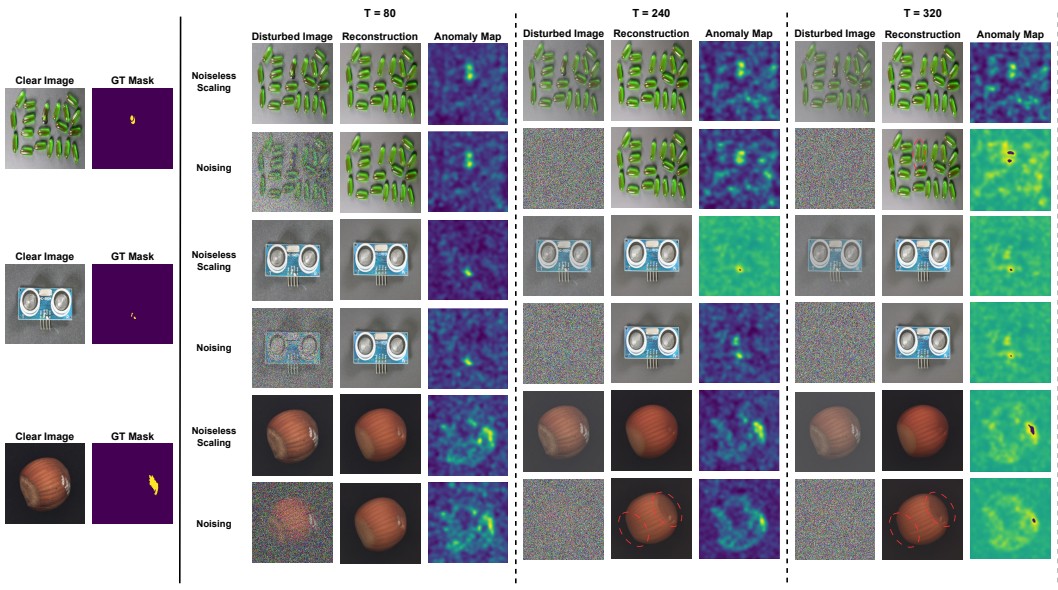

Figure 15: Impact of noiseless scaling versus noising on reconstruction and anomaly map construction on Categories Capsules and PCB1 of VisA and Hazelnut of MVTec. Failed reconstructions are circled in red. The disturbed image level columns are only added for visualization, D3AD performs scaling/noising on the latent level.

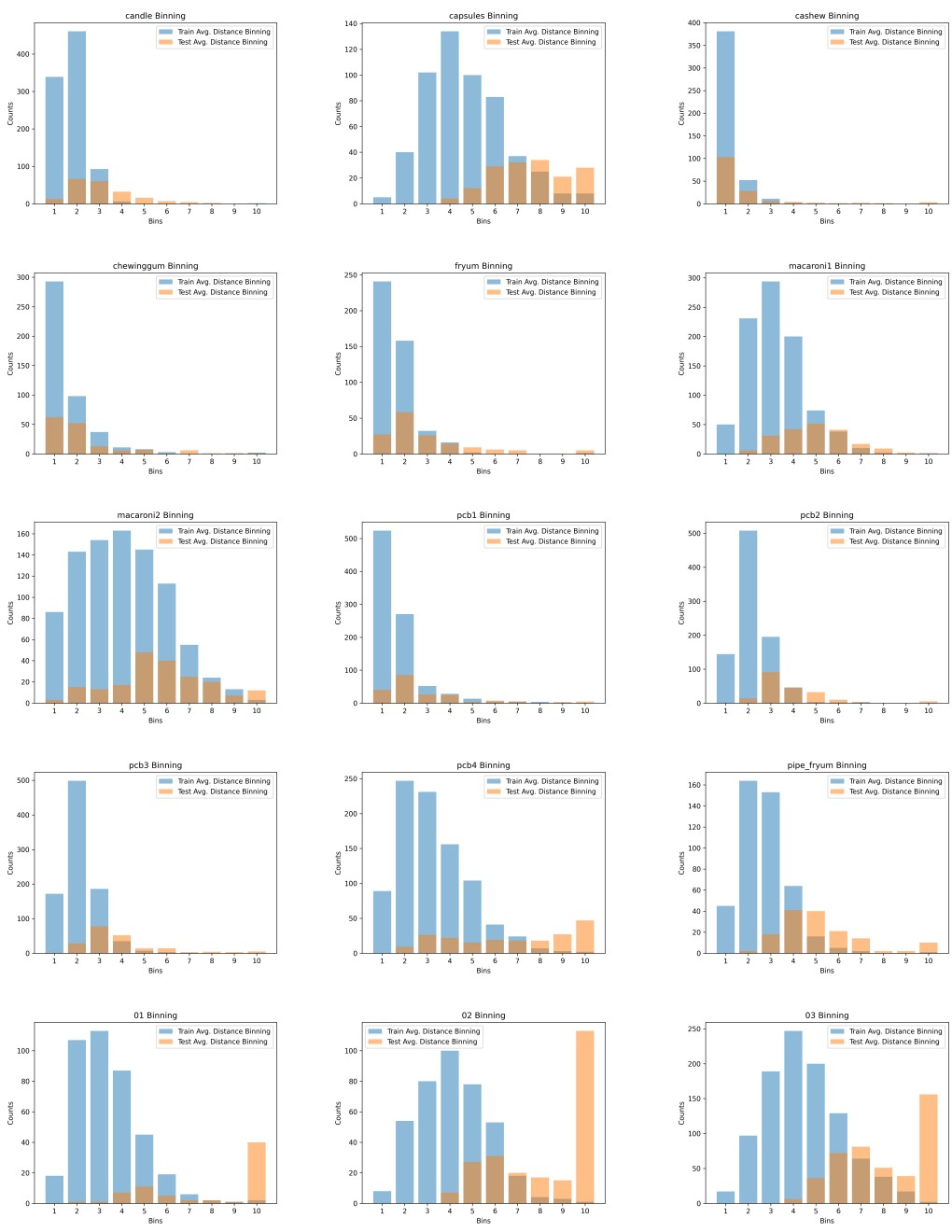

Figure 16: Binning distributions for the training and test set for all categories of the VisA and BTAD benchmark.

