# OpenReview forum: "D3AD: DYNAMIC DENOISING DIFFUSION PROBABILISTIC MODEL FOR ANOMALY DETECTION"
_ICLR.cc/2024/Conference — Submitted to ICLR 2024_

### Official Review · Reviewer_vjSv · 2023-10-16

**Soundness:** 1 poor
**Presentation:** 2 fair
**Contribution:** 1 poor
**Rating:** 3
**Confidence:** 5

**Summary:**

The paper proposes D3AD, an anomaly detection method based on the pre-trained diffusion model. Authors propose two components that deviate from the conventional works; dynamic implicit conditioning which selects dynamic time steps for noising and guidance based on the reference latent that is not constructed by adding noise. Such contributions are integrated with domain adaptation strategy and final anomaly map reconstruction to consist of D3AD.  Authors experiment with D3AD in the VisA and BTAD datasets.

**Strengths:**

(1) The idea of dynamic time for anomaly detection is interesting.

**Weaknesses:**

The paper has various issues on soundness and significance. Specifically, I am confused about why the proposed method should work, given that the method comprises differences against existing diffusion-based anomaly detection. Further, unlike the authors insist, D3AD misses various benchmarks, competitive baselines, and even metrics for comparison. Below are my specific details on the issue I have on this paper.


**Originality & Soundness & Clarity** \
(1) The paper overclaims about the domain adaptation technique. DDAD [1] also uses a similar objective to fine-tune the feature extractor. I would rather appreciate that the authors lower the tone.\
(2) The authors formulate reference $z_{\hat{T}}$ as a multiplication of the latent without any noise addition. However, I do not find any support for why such guidance on the latent should work. The authors may elaborate on why this helps in detail via further analysis.\
(3) I do not understand how the binary search function $\psi$ can choose the right time on the bin. Further details should be provided on how this works or at least introduce related concepts.\
(4) I am also confused about how the method constitutes a "predefined histogram" in Figure 3.

**Significance**\
(1) The authors claim their methods show **state-of-the-art** performance but I doubt this is true. For example, [2] scores I-AUC of 97.6, P-AUC of 98.4, and PRO of 94.9 while this method scores 96.0/97.9/94.1. I don't feel that this method outperforms DDAD even since DDAD scores 99.3 I-AUC in the VisA dataset, which is somehow not shown in Table 1. The authors should compare all the baselines in previous conferences (e.g. ICML, CVPR, ICCV) to support this bold claim.\
(2) I also feel like three metrics least (I-AUROC, P-AUROC, p_PRO) should be compared as a whole in the major table (e.g. Table 1).\
(3) I am curious why the authors dropped the MvTec dataset for comparison since most anomaly detection algorithms are compared in the dataset. It is hard to assert that the method shows state-of-the-art performance without comparing the MvTecAD dataset in my opinion.\
(4) Even addressing this issue, I am confused about the gain of this paper since the proposed model uses a latent diffusion model, unlike the other diffusion-based methods.\
(5) Furthermore, diffusion-based methods show slow sampling speed due to computation-heavy U-Nets and multiple inference steps. It would be beneficial to note the computational cost of this method compared to competitive baselines.\



Overall, I do not find defining controllable operations in the latent space explicitly sound. Given such confusion and the excessive computational cost of diffusion-based anomaly detection methods, the paper's performance is not that competitive. This can be observed by simple baselines (e.g. PatchCore shows better P-AUROC in the VisA dataset) outperforming D3AD in some metrics while the most competitive benchmark has not been featured for comparison. This as a whole makes me skeptical of how this idea is surprising or valid as a competitive anomaly detection method. Hence, I lean for rejection.

**Questions:**

see Weakness


**References**\
[1] Anomaly detection with conditioned denoising diffusion models, arXiv 2023\
[2] Remembering normality: memory-guided knowledge distillation for unsupervised anomaly detection, ICCV 2023

---

> ### Author Response · Authors · 2023-11-16
>
> Thank you for your constructive feedback. We are prepared to promptly address and incorporate the necessary updates in the paper to resolve your concerns. We would like to offer a short correction that we don’t use a pretrained diffusion model but rather train a latent diffusion model on each category.
>
> $\\\\$
>
> $\\textbf{Originality Soundness Clarity}$
>
> $\\textbf{W1:}$ Our domain adaptation approach differs from DDAD[1]. Unlike DDAD, which minimizes a cosine distance between the reconstruction and a randomly sampled training example, our objective function minimizes a cosine distance between the extracted features of an input images and its reconstructions. This not only learns intricacies of the target domain but also mitigates artifacts arising from using a pre-trained VAE for encoding and decoding images into latent.
>
> $\\textbf{W2:}$ We've conducted comprehensive analytical experiments, now detailed in the appendix (Fig 14), to empirically support our claim. Notably, our model applies noising and denoising to latents, which are inherently robust compared to pure images. However, surprisingly, even with no noise and just scaling, latents are perturbed sufficiently for faithful reconstruction, underscoring their resilience, see appendix figures 7 and 8.
>
> $\\textbf{W3:}$ Thank you for pointing out these concerns. We've revised the explanation of binary search function and bin construction in section 4.1 to provide clarity. The use of binary search denotes mapping the average distance of a datapoint to its k-neighbours into the fitting bin. For instance, an average distance of 4.5 would be placed in bin 1 which is defined to cover the range (1,5].
>
> $\\textbf{W4:}$ We have updated the wording from ‘predefined histogram’ to ‘predefined bins’.
>
> $\\\\$
>
> $\\textbf{Significance}$
>
> $\\textbf{W1:}$ We appreciate your observation. Our comparison in Table 1 includes all published methods available up to the submission deadline. [2] was published after this cutoff, and as such, not featured, therefore we maintain SOTA performance. Our segmentation performance, Table 2, is intentionally confined to relevant diffusion-based methods to offer additional insight showing SOTA segmentation performance. We have now augmented this table with I-AUROC.
>
> $\\textbf{W2:}$ As detailed in our methods section, we chose to emphasize the PRO metric in Table 1. This decision is based on PRO's ability to treat anomalies across scales equally, showcasing our approach's efficacy in handling a variety of anomalies, figures 1, 7 and 10. We have further augmented the appendix with Pixel AUROC in table 7 as requested.
>
> $\\textbf{W3:}$ We addressed this in the general response, here a short summary: We focused on the VisA[3] dataset as it is more challenging however, we now performed a preliminary study on MVTec[4] which is detailed in the appendix. Comparison to other reconstruction-based methods is provided in the general response. We are pleased to report that our method demonstrates comparable performance out-of-the-box.
>
> $\\textbf{W4:}$ The utilization of latent diffusion models for anomaly detection is a novel and advantageous aspect of our approach. This choice is motivated by computational efficiency compared to standard diffusion models. Latents, being a more robust representation than raw images, play a pivotal role in enabling the reconstruction of large missing components, figure 7, while mitigating background artifacts which is a common challenge in reconstruction-based models. Furthermore, our introduction of a dynamic step mechanism is the primary contribution, addressing a limitation of standard diffusion models.
>
> $\\textbf{W5:}$ We acknowledge your point and have addressed it by including inference times in the appendix. We are pleased to share that our method demonstrates competitive inference times compared to representation-based methods.
>
>
> $\\\\$
>
> We trust these clarifications enhance the understanding of our methodology.
>
> $\\\\$
>
>
> [1] Mousakhan, Arian, Thomas Brox, and Jawad Tayyub. "Anomaly Detection with Conditioned Denoising Diffusion Models." arXiv preprint arXiv:2305.15956 (2023).
>
> [2] Gu, Zhihao, et al. "Remembering Normality: Memory-guided Knowledge Distillation for Unsupervised Anomaly Detection." Proceedings of the IEEE/CVF International Conference on Computer Vision. 2023.
>
> [3] Zou, Yang, et al. "Spot-the-difference self-supervised pre-training for anomaly detection and segmentation." European Conference on Computer Vision. Cham: Springer Nature Switzerland, 2022.
>
> [4] Bergmann, Paul, et al. "MVTec AD--A comprehensive real-world dataset for unsupervised anomaly detection." Proceedings of the IEEE/CVF conference on computer vision and pattern recognition. 2019.

---

> ### Comment · Reviewer_vjSv · 2023-11-19
>
> Thank you for your response.
> While some of my concerns are resolved, my main concerns on the empirical significance and the novelty are not resolved. I think the paper is an extension of existing ideas on the previous methods, except extending it from image to latent. While the perception of novelty can be pretty subjective, this paper fails to be a competitive baseline method. In my opinion, the paper should improve the result through several rounds of submission. Below are my detailed comments.
>
> Original Soundness Clarity\
> W1: As mentioned above, I do not buy the domain adaptation method being novel since it is just an extension of pixel-based distance to feature-based distance, and how many previous works have been in such an idea (e.g. reverse distillation-based methods)\
> W2: Figure 14 is perplexing. There is no guarantee still why the scaling works, only empirical results (which are not state-of-the-art). Figure 14 seems to show that additive noise degrades the result. There are millions of bad methods and comparing them does not show any idea why the proposed scaling without additional noise works.
>
> Significance\
> W1: I still do not think Table 1 contains all competitive methods. I want to ask why the authors did not include DDAD and DiffusionAD in Table 1 although they included them in Table 2. Furthermore, I can see many papers that outperform D3AD on the MvTecAD and BTAD datasets. For example, see [1] and [2] for examples of competitive baselines (I'm also interested why the authors included [2] in the Appendix of the paper but not in the general response)\
> W2: I feel the paper overemphasizes the importance of PRO against others. That's a debatable view. Better Image-AUROC performance is also a valuable metric and I do not agree that the method excelling only on the PRO metric doesn't guarantee its efficacy in anomaly detection.\
> W3: The same idea on W2 also applies here. I do agree that MvTec is a heavily tested dataset, but it also contains various competitive methods only experimented with there. VisA has fewer experimented methods and it is hard to extend the idea of the method being competitive only compared to the VisA dataset.\
> W5: There are much faster methods with better inference speeds (ex [2] shows 6x faster than PatchCore and the method is slower than PatchCore while authors regard them as "competitive")
>
> Overall, I feel the survey of competitive baselines is further required in the paper. Furthermore, I think the authors should clarify the survey of existing methods. Also, this method only scores competitive performance in 1 metric out of 3 and 1 dataset out of 3. I still lean to rejection.
>
> p.s.) For my clarification about the "pre-trained", I did not find any differences in D3AD's training scheme on LDM against other papers that are based on the latent diffusion model. Hence, this can be seen as pre-trained.
>
> **References**
>
>
> [1] PNI: Industrial Anomaly Detection using Position and Neighborhood
> Information [ICCV 2023] (link: https://arxiv.org/pdf/2211.12634.pdf)
>
> [2] Revisiting Reverse Distillation for Anomaly Detection [CVPR 2023]

---

> > ### Author Response · Authors · 2023-11-22
> >
> > Original Soundness Clarity
> >
> > Thank you for your swift feedback. Your input has helped improve our work. While we recognize that anomaly detection has recently been researched with diffusion models, our primary contribution lies in introducing dynamic noising time steps—a novel aspect not explored in existing literature within this framework. We are keen to learn of any prior work that has ventured into a comparable task.
> >
> > Regarding figure 14 we have added an additional Figure 15 in response to reviewer xkrq. This figure demonstrates a noising vs scaling paradigm along with reconstructions from each. We hope that this resolved the ambiguity as it clearly demonstrates superior reconstruction achieved with scaling only and hallucinations generated with noising at the same time step. We have offered a discussion on this behaviour is Appendix A2.1.
> >
> > Significance
> >
> > In Table 1, we present a comparison with papers that were peer-reviewed as of the submission deadline. Notably, at that time, diffusion-based papers were largely under submission, with the sole exception being AnoDDPM. To address this, we compiled an additional table dedicated to diffusion-based methods, offering a comprehensive comparison against such highly related approaches. We included in the general response MVTec results with only reconstruction-based methods. [2] does not qualify as reconstruction-based, while, as mentioned in our original response, [1] is ICCV2023 which supersedes the deadline of ICLR.
> >
> > Similar to [3], our primary goal is to enhance segmentation performance, measured by PRO, particularly focusing on anomalies of diverse scales. We note that our work ties with the state-of-the-art on the VISA dataset in I-AUROC while exceeding in PRO among peer-reviewed studies. Furthermore, as evident by MVTec results above D3AD is superior amongst reconstruction-based results.
> >
> > In summary, our study encompasses all peer-reviewed baselines available to us, including emerging, not-yet peer-reviewed diffusion-based approaches. In our analyses, when compared only to peer-reviewed reconstruction-based methods, our performance is superior across all three metrics on VisA and two of three on both MVTec and BTAD, underscoring the robustness and relevance of our findings. Moreover, we demonstrate competitive results in two of three metrics on the VisA benchmark and one of three on the BTAD benchmark amongst all peer-reviewed baselines.
> >
> > [1] Bae, Jaehyeok, Jae-Han Lee, and Seyun Kim. "Pni: industrial anomaly detection using position and neighborhood information." Proceedings of the IEEE/CVF International Conference on Computer Vision. 2023.
> >
> > [2] Tien, Tran Dinh, et al. "Revisiting reverse distillation for anomaly detection." Proceedings of the IEEE/CVF Conference on Computer Vision and Pattern Recognition. 2023.
> >
> > [3] Lu, Fanbin, et al. "Removing anomalies as noises for industrial defect localization." Proceedings of the IEEE/CVF International Conference on Computer Vision. 2023.

---

### Official Review · Reviewer_Emgj · 2023-10-30

**Soundness:** 2 fair
**Presentation:** 1 poor
**Contribution:** 2 fair
**Rating:** 3
**Confidence:** 5

**Summary:**

This paper presents a new framework called D3AD that aims to resolve the issue of localizing anomalies of varying scales using diffusion models. The framework enhances the capability of diffusion models by integrating a dynamic conditioning mechanism with prior information from a KNN model, a domain adaptation mechanism, and a specialized latent diffusion model. The proposed method is supported by thorough evaluation and analysis and demonstrates superior performance on two benchmark datasets.

**Strengths:**

1) The paper introduces novel contributions to the field of anomaly detection using diffusion models. including dynamic conditioning, noiseless input scaling, and latent diffusion model.
2) The proposed method achieves state-of-the-art performance on two benchmark datasets, VisA and BTAD, demonstrating its effectiveness in anomaly detection and localization.

**Weaknesses:**

1) The paper provides a well-structured framework, but the explanation of each component lacks clarity and precision. In addition, the paper lacks verification of the validity of some arguments.
2) The description of some methods used in the paper is obscure. For example, the paper does not provide a clear and rigorous mathematical explanation of the proposed DIC method, but only a short description of it.
3) Some of experimental results in the paper lack detailed discussions. For example, the paper lacks careful analysis of qualitative results and does not link experimental results to the proposed methods logically.
4) The paper is imprecise and unpolished. There are a number of grammatical errors in the text, so careful checking and revision is recommended.

**Questions:**

There are some key details missed so that it is difficult to draw convincing conclusions:
1)	How did you determine an noiseless and only scaled input $x_{\hat T}$ is optimal for faithful reconstruction?
2)	Why did you chose |B|=10?
3)	What is meaning of “unnoised”? It would be better to replace “unnoised” with “noiseless”.
4)	Could you please elaborate the qualitative effect of DIC based on Figure 6?
5)	How did you localize and detect anomalies of varying scales? The main body lacks a detailed description of the solution to this problem and the corresponding experimental analyses.

---

> ### Author Response · Authors · 2023-11-16
>
> Thank you for providing your feedback, we would greatly appreciate to receive specific points and further clarification regarding the weaknesses you have highlighted.
>
> $\\textbf{W1:}$ We respectfully disagree as we adhere to standard convention in notation from well-established diffusion model literature. We are unable to identify specific arguments which lack validity. Could you please specify these points for further clarification?
>
> $\\textbf{W2:}$ To address this concern, could you please specify which part of our DIC method explanation you find obscure? We are open to understanding your expectations for further rigor.
>
> $\\textbf{W3:}$ Due to space constraints, an in-depth discussion had not been included, however, in light of your feedback, we've added more qualitative figures 7 and 8 in the appendix material as well as a discussion on linking results to the various components of the framework.
>
> $\\textbf{W4:}$ Once again, we're unable to pinpoint specific shortcomings in the writing. We would greatly appreciate your guidance in identifying precise areas of concern to address and improve.
>
>
> $\\textbf{Q1:}$ We've conducted comprehensive analytical experiments, now detailed in the appendix (Fig 14), to empirically support our claim. Notably, our model applies noising and denoising to latents, which are inherently robust compared to pure images. However, surprisingly, even with no noise and just scaling, latents are perturbed sufficiently for faithful reconstruction, underscoring their resilience, see appendix figures 7 and 8.
>
> $\\textbf{Q2:}$ The choice was made empirically, and an ablation will be added in camera ready.
>
> $\\textbf{Q3:}$ Thank you for the suggestion, we have reworded to ‘noiseless’.
>
> $\\textbf{Q4:}$ Figure 6 demonstrates that a dynamically computed time step (DIC Mask) provides the most similar anomaly prediction mask to the ground truth (GT) in comparison to masks from fixed noising time steps as shown in remaining sequence -- from 100\% to 25\% of $T=80$. We have updated this description in main paper.
>
> $\\textbf{Q5:}$ As described in the methods section and illustrated in figure 2, we first estimate the anomaly severity by initially using a KNN to compare a test image to the training data. This severity is translated into the scaling level needed for perturbing the latent of the test image. The diffusion model then reconstructs the scaled (and noiseless) latent which is decoded into an image reconstruction. A comparison of the input and reconstruction results in localisation of anomalies of varying severity.

---

### Official Review · Reviewer_xkrq · 2023-11-02

**Soundness:** 2 fair
**Presentation:** 2 fair
**Contribution:** 2 fair
**Rating:** 3
**Confidence:** 5

**Summary:**

This paper proposes an approach termed dynamic denoising diffusion probabilistic model (D3AD) for the task of anomaly detection. D3AD designs a pipeline with three specific components: (i) a dynamic conditioning mechanism based on KNN, (ii) a domain adaptation mechanism that aims to learn the representation in the target domain, (iii) a latent diffusion model for reconstruction. D3AD achieves SOTA performance on some metrics in two datasets.

**Strengths:**

•	This paper proposes a novel dynamic noise injection mechanism that makes sense.

•	 The performance gains for anomaly localization look good, especially on PRO metric.

**Weaknesses:**

•	D3AD has not been validated on the most widely used dataset, MVTec[1], and I'm curious about its performance on this dataset.

•	The impact of the dynamic noise amount on the reconstruction results is not clear from this paper, and it appears that the reconstruction quality is not very good based on Figure 3.

•	The novelty of this work is very limited. The proposed conditioned denoising and domain adaptation are similar to DDAD[2]. Besides, the anomaly scoring paradigm is similar to RD4AD[3]. As a result, the framework of D3AD looks like a combination of existing works.

•	D3AD performs poorly on Image AUROC compared to the diffusion-based methods mentioned in the paper, such as DDAD and DiffusionAD.

[1] MVTec AD--A comprehensive real-world dataset for unsupervised anomaly detection. Bergmann P, Fauser M, Sattlegger D, et al.CVPR 2019.

[2] Anomaly Detection with Conditioned Denoising Diffusion Models. Mousakhan A, Brox T, Tayyub J. arXiv 2023.

[3] Anomaly detection via reverse distillation from one-class embedding. Deng H, Li X. CVPR, 2022.

**Questions:**

•	Is there more detailed experimentation or theoretical basis to explain why noise is not needed during inference?

•	Are there more visual results showcasing the reconstruction of different types of anomaly regions?

•	How does D3AD perform on MVTec, which is one of the most widely used datasets?

---

> ### Author Response · Authors · 2023-11-16
>
> Thank you for your valuable comments, we aim to address all points and incorporate all feedback.
>
>  $\textbf{W1:}$ We addressed this in the general response, here a short summary: We focused on the VisA[1] dataset as it is more challenging however, we now performed a preliminary study on MVTec[2] which is detailed in the appendix. Comparison to other reconstruction-based methods is provided in the general response. We are pleased to report that our method demonstrates comparable performance out-of-the-box.
>
> $\textbf{W2:}$ We addressed this in general response, here a short summary: We've updated figure 3 and introduced two new figures 7 and 8 in the appendix which clearly showcases compelling reconstructions achieved with our Dynamic Implicit Conditioning (DIC) mechanism. Figure 8 present a qualitative comparison of reconstruction against current SOTA in reconstruction-based method DRAEM [3] and shows significant superiority. These enhancements emphasize the robustness and effectiveness of DIC in diverse scenarios.
>
> $\textbf{W3:}$ While we leverage DDAD's [4] conditioning mechanism, we don't claim novelty in this regard. Our contribution lies in extending this mechanism by introducing dynamic time step computation based on an early estimate of an anomaly’s severity using k-NN. Additionally, our domain adaptation approach differs as well. Unlike DDAD, which minimizes a cosine distance between the reconstruction and a randomly sampled training example, our objective function minimizes a cosine distance between the extracted features of an input images and its reconstructions.
>
> Furthermore, our method diverges significantly from RD4AD[5] in anomaly scoring. RD4AD employs a teacher encoder and a student decoder and measures the distance between the representations from these two blocks. Our feature distance $f_{map}$, in contrast, computes a distance between the feature representations (extracted from an encoder) of an input image and its reconstruction, see figure 4. Additionally, our approach deviates from the conventional teacher-student model by introducing a second distance measure $l_{map}$. This measures the disparity between the latent z and its reconstruction z_hat generated by the diffusion model.
>
> $\\textbf{W4:}$ Our primary focus is on the segmentation problem, addressing varying degrees of anomaly severity. The design of D3AD emphasizes achieving high PRO scores, aligned to recent work [6]. We've now also included Pixel AUROC in the appendix for completeness. Our method's strength lies in its segmentation capabilities, and we believe the appendix results will provide a more comprehensive understanding of its performance.
>
> [1] Zou, Yang, et al. "Spot-the-difference self-supervised pre-training for anomaly detection and segmentation." European Conference on Computer Vision. Cham: Springer Nature Switzerland, 2022.
>
> [2] Bergmann, Paul, et al. "MVTec AD--A comprehensive real-world dataset for unsupervised anomaly detection." Proceedings of the IEEE/CVF conference on computer vision and pattern recognition. 2019.
>
> [3]Zavrtanik, Vitjan, Matej Kristan, and Danijel Skočaj. "Draem-a discriminatively trained reconstruction embedding for surface anomaly detection." Proceedings of the IEEE/CVF International Conference on Computer Vision. 2021.
>
> [4] Mousakhan, Arian, Thomas Brox, and Jawad Tayyub. "Anomaly Detection with Conditioned Denoising Diffusion Models." arXiv preprint arXiv:2305.15956 (2023).
>
> [5] Deng, Hanqiu, and Xingyu Li. "Anomaly detection via reverse distillation from one-class embedding." Proceedings of the IEEE/CVF Conference on Computer Vision and Pattern Recognition. 2022.
>
> [6] Lu, Fanbin, et al. "Removing Anomalies as Noises for Industrial Defect Localization." Proceedings of the IEEE/CVF International Conference on Computer Vision. 2023.

---

> > ### Comment · Reviewer_xkrq · 2023-11-20
> >
> > •	The performance of D3AD is limited; it significantly lags behind representation-based methods on the MVTec dataset, with marginal improvements on VisA and BTAD datasets. Consequently, if both the performance and speed of D3AD fall behind those of representation-based methods, it does not necessarily get prioritized in practical applications.
> >
> > •	The enhancement brought about by the noise-less paradigm remains ambiguous. Figures 11 and 12 solely compare differences in forward results, without a comparative analysis of variations in reconstruction and anomaly detection results. This fails to provide a clear, intuitive demonstration.
> >
> > •	The visual results in Figure 7 do not distinctly illustrate the dynamic noising step of D3AD. It would be more informative to present, for various anomaly types, the specific dynamic noising steps chosen by D3AD in the figure. In addition, these results should be compared with cases where DDAD or D3AD uses a fixed step, demonstrating differences in reconstruction and anomaly detection results. This will provide a more straightforward illustration of how the dynamic noising step enhances performance.
> >
> > Based on the aforementioned considerations, I am inclined towards recommending rejection.

---

> ### Author Response · Authors · 2023-11-22
>
> Thank you for your constructive feedback. It helped us to improve our paper significantly.
>
> $\\textbf{A1:}$ We agree that representation-based methods are generally superior to reconstruction-based approaches, however, in our experience, practical applications favour reconstruction-based method as they generate a flawless version of the defected input. This aids in 1) gaining insight into the model’s reasoning and thereby instilling trust and 2) guiding inexperienced workers to potentially rectify the defect that has been identified. We obtain superior segmentation accuracy on MVTec amongst other reconstruction-based methods (table above) and report highest performance on VISA amongst both reconstruction and representation-based method. We believe that our work still presents an interesting advancement amongst reconstruction-based diffusion methods which has been underexplored in literature.
>
> $\\textbf{A2:}$ Thank you for highlighting this ambiguity. We have taken your suggestions into account and composed a new Figure 15 which presents noising vs scaling paradigm on different perturbance levels. This clarifies the impact of scaling vs noising as seen in reconstructions and consequently anomaly maps. Note the capsules category row 2, T = 320, noising results in higher degradation of the image resulting in a reconstruction which hallucinates an additional green capsule (red circle), whereas an only scaled image retains more of the input image while perturbing pixels enough for a faithful reconstruction in comparison.
>
> $\\textbf{A3:}$ We appreciate this feedback and revised Figure 7 to now feature an additional column stating the percentage disturbance level chosen by DIC. It is now clear that since larger anomalies, rows 1,3-5, need a higher perturbation, DIC correctly selects a high value whereas finer defects such as blemishes, result in a lower disturbance level.

---

### Author Response · Authors · 2023-11-16
**General Response**

We thank all reviewers for their valuable feedback. All reviewers found our approach intriguing, and particularly admired the dynamic step conditioning. Reviewers xkrq and Emgj also found our approach novel and commended the state-of-the-art anomaly localisation and detection results on two datasets. We address common concerns below and offer specific answers to questions per reviewer in the responses.

$\\textbf{Impact of DIC:}$ Thank you for pointing out the need for clarity on the impact of dynamic noise in perceived reconstruction in Figure 3. We've addressed this concern by updating figure 3 and introducing two new figures 7 and 8 in appendix A which clearly showcases compelling reconstructions achieved with our Dynamic Implicit Conditioning (DIC) mechanism. Figure 7, rows 1, 2 and 6-9 demonstrate flawless reconstruction of minor defects, while rows 3, 4, and 5 showcase the ability to handle entire missing or majorly flawed components. Row 3 is of particular importance as it showcases reconstruction of a complete component. Figure 8 present a qualitative comparison of reconstruction against current SOTA in reconstruction-based methods DRAEM [1] and show significant superiority. These enhancements emphasize the robustness and effectiveness of DIC in diverse scenarios. We believe these additions effectively address your concerns.

$\\textbf{MVTec[2] Dataset:}$ We thank the reviewers for highlighting the MVTec dataset. We intentionally focused on the VisA[3] dataset to tackle a more challenging task of having multiple items per image and complex structures such as PCBs. We consciously decided not to address MVTec, which is performance-saturated, making it challenging to showcase our approach that is designed for varying scales of anomalies. However, in response, we conducted a preliminary study on MVTec without hyperparameter tuning and pleased to report that our method demonstrates reputable performance out-of-the-box  achieving SOTA segmentation accuracy amongst other reconstruction-based methods. Complete results along with qualitative figures have been included in Appendix A Table 6, Figures 7 and 8.


\\begin{array}{ccccc}
\\textbf{Method}& GANomaly  &SkipGANomaly  & DRAEM   &D3AD (Ours)\\\\
\\hline \\\\
\\textbf{Carpet} &(20.3,-)  &(70.9,-) &(\\textbf{97.0},95.5,92.9)  & (94.2,\\textbf{97.6},\\textbf{95.1}) \\\\
\\textbf{Grid} & ( 40.4,-)  &(47.7,-) &(99.9,\\textbf{99.7},\\textbf{98.4})&(\\textbf{100},99.2,96.9) \\\\
\\textbf{Leather} &( 41.3,-)   &(60.9,-)  &(\\textbf{100},98.6,98.0) & (98.5,\\textbf{99.4},\\textbf{98.1}) \\\\
\\textbf{Tile} &(40.8 ,-)   &(29.9,-)  &(\\textbf{99.6},\\textbf{99.2},\\textbf{98.9}) &(95.5,94.7,93.6) \\\\
\\textbf{Wood}    &( 74.4,-)  &(19.9,-)  &(99.1,\\textbf{96.4},\\textbf{94.6}) & (\\textbf{99.7},95.9,91.0) \\\\
\\textbf{Bottle} &(25.1,-)  &(85.2,-)  &(99.2,\\textbf{99.1},\\textbf{97.2}) & (\\textbf{100},98.6,96.0)  \\\\
\\textbf{Cable} &( 45.7,-)   &(54.4,-)  &(91.8,\\textbf{94.7},76.0)  &(\\textbf{97.8},93.3,\\textbf{87.3})  \\\\
\\textbf{Capsule}&( 68.2,-) &(54.3,-)  &(\\textbf{98.5},94.3,\\textbf{91.7})  & (96.6,\\textbf{97.9},90.7) \\\\
\\textbf{Hazelnut} &( 53.7,-)   &(24.5,-)  &(\\textbf{100},\\textbf{99.7},\\textbf{98.1})& (98.0,98.8,91.8) \\\\
\\textbf{Metal nut}  &(27.0 ,-)   &(81.4,-)  &(98.7,\\textbf{99.5},\\textbf{94.1}) & (\\textbf{98.9},96.1,89.7) \\\\
\\textbf{Pill}  &( 47.2,-)  &(67.1,-) &(98.9,97.6,88.9) & (\\textbf{99.2},\\textbf{98.2},\\textbf{96.2}) \\\\
\\textbf{Screw} &( 23.1,-)   &(87.9,-)  &(\\textbf{93.9},97.6,98.2)  & (83.9,\\textbf{99.0},\\textbf{95.5}) \\\\
\\textbf{Toothbrush} &(37.2 ,-)   &(58.6,-)  &(\\textbf{100},98.1,90.3)& (\\textbf{100},\\textbf{99.0},\\textbf{94.6}) \\\\
\\textbf{Transistor}&( 44.0,-)   &(84.5,-)  &(93.1,90.9,81.6) & (\\textbf{96.8},\\textbf{95.6},\\textbf{86.9})  \\\\
\\textbf{Zipper}  &(43.4 ,-)   &(76.1,-)  &(\\textbf{100},\\textbf{98.8},\\textbf{96.3}) & (98.2,98.3,95.3)\\\\
\\hline
\\textbf{Average}&(42.1 ,-)  &(60.2,-)  &(\\textbf{98.0},97.3,93.0)  & (97.2,\\textbf{97.4},\\textbf{93.3})\\\\
\\end{array}

[1]Zavrtanik, Vitjan, Matej Kristan, and Danijel Skočaj. "Draem-a discriminatively trained reconstruction embedding for surface anomaly detection." Proceedings of the IEEE/CVF International Conference on Computer Vision. 2021.

[2] Bergmann, Paul, et al. "MVTec AD--A comprehensive real-world dataset for unsupervised anomaly detection." Proceedings of the IEEE/CVF conference on computer vision and pattern recognition. 2019.

[3] Zou, Yang, et al. "Spot-the-difference self-supervised pre-training for anomaly detection and segmentation." European Conference on Computer Vision. Cham: Springer Nature Switzerland, 2022.

---

### Meta-Review · Area_Chair_umi1 · 2023-12-04

**Metareview:**

The reviewers were unanimous in their vote to reject, feeling that the submission was not ready for publication. Issues raised include lack of clarity in the description of the (potentially an overly complex) technical approach.

**Justification For Why Not Higher Score:**

Reviewers do not support acceptance.

**Justification For Why Not Lower Score:**

N/A

---

### Decision · Program_Chairs · 2024-01-16

Reject